# ERPV: Enhancing Visual Reinforcement Learning with Partially Reliable Knowledge from VLMs

## Abstract

Visual Reinforcement Learning (VRL) aims to learn optimal control policies from scratch, a process that often suffers from low exploration efficiency. Integrating large-scale vision-language models (VLMs) offers a promising solution, as they provide rich prior knowledge about the environment. However, VLMs are only *partially reliable* when directly applied to VRL: the inferred actions may be wrong in certain states, and the inability to identify reliable action alignment can result in excessive exploration by the agent. We propose ERPV, a novel method that effectively enhances VRL with partially reliable knowledge from VLMs. ERPV introduces two key modules: (1) Value-aware Policy Guidance, which estimates the reliability of VLMs across different states and adaptively selects trustworthy VLM-inferred actions to guide policy learning; (2) VLMs-guided Entropy Regularization, which reduces over-exploration by comparing the confidence between VRL policy and VLMs-inferred actions. Extensive experiments show that, compared to the state of the art, ERPV achieves competitive performance in both policy effectiveness and sample efficiency under diverse, complex visual control tasks. The code has been placed in the supplementary materials.

## 1 Introduction

Visual reinforcement learning (RL) has achieved significant success in game playing (Scheller et al., 2020; Lin et al., 2021), robotic control (Noh & Myung, 2022; Huang et al., 2023), and computer graphics (Wu et al., 2022; Chowdhury et al., 2024). However, visual RL typically requires extensive environment interactions to learn policies, leading to low exploration efficiency. This limits its applicability in complex scenarios where sample efficiency is critical (Yu, 2018; D'Oro et al., 2022). One solution is offline reinforcement learning (Shi et al., 2021; Tianci et al., 2024), which leverages pre-collected trajectories but depends on large amounts of high-quality expert data.

In recent years, large models (LMs), including vision-language models (VLMs) and large language models (LLMs), have shown strong capabilities in understanding and reasoning on various tasks (Guo et al., 2024; Dorka et al., 2024; Hu & Sadigh, 2023; Wang et al., 2024c; Zhang et al., 2024a). However, their slow inference speed, high memory usage, and sensitivity to prompts (Gandhi & Gandhi, 2025) hinder their use in real-time scenarios. In this work, we instead consider transferring zero-shot knowledge from pre-trained VLMs to visual RL agents. VLMs provide only training guidance and do not interact with the environment. Moreover, they are excluded during testing, ensuring real-time performance.

Recent studies have explored how to leverage the general knowledge embedded in LMs to enhance RL. For example, some methods empoly LMs to extract state features (Mezghani et al., 2023; Grigsby et al., 2023; Paischer et al., 2022) or design rewards (Ma et al., 2023; Peng et al., 2025). While beneficial for policy learning, they do not directly improve exploration. Others employed LMs as actors to exploit prior knowledge (Li et al., 2022; Carta et al., 2023; Tan et al., 2024; Yao et al., 2023; Peiyuan et al., 2025; Shi et al., 2023; Lee et al., 2025), at the cost of high computational overheads. To mitigate this issue, recent approaches adopt a teacher–student paradigm, where LMs act as teachers that guide lightweight policy networks (Zhou et al., 2024; Xu et al., 2025a; Liu et al., 2024; Wu et al., 2024). These methods assume strong zero-shot performance from LMs,

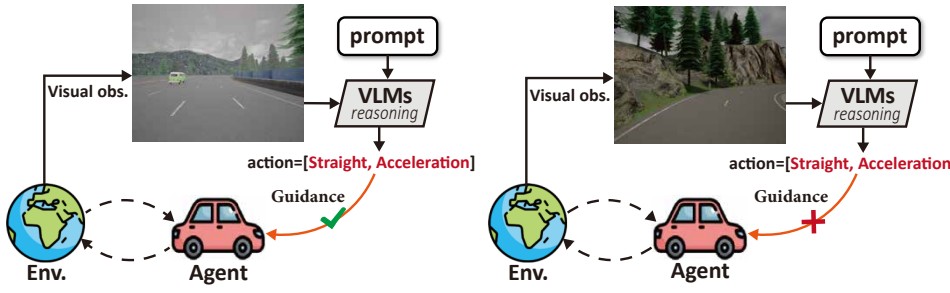

(a) Reliable VLMs reasoning.                    (b) Unreliable VLMs reasoning.

Figure 1: A reliability analysis of VLM inference in CARLA (Dosovitskiy et al., 2017) using Qwen2-VL-7B (Wang et al., 2024a) shows two distinct patterns: (a) In straight lanes with safe vehicle distance, the agent maximizes rewards through acceleration. Here, VLM-generated reasoning actions are reliable and effectively guide agent learning. (b) In right-turn lanes, however, limited task-specific training and ambiguity in grounding (Zhang et al., 2023) lead to unreliable VLM inferences, which can result in suboptimal learning if used directly.

yet overlook their limitations. Indeed, VLMs may suffer from hallucination and ambiguity (Shahgir et al., 2024; Zhang et al., 2023), leading to unreliable inferences and potential biases. Such issues ultimately hinder policy learning in visual RL, as illustrated in Fig. 1.

To this end, we propose a method to enhance visual RL by utilizing partially reliable knowledge from VLMs (ERPV). The first issue is to estimate the reliability of VLMs across different states. To address this, we introduce a Value-aware Policy Guidance (VPG) module. VPG employs a pre-trained critic network to evaluate the state–action values (i.e., $Q$ values) of actions inferred by VLMs. These $Q$ values are compared with those of the current RL policy to compute the relative advantages of VLM-inferred actions, which serve as a policy guidance factor. This dynamic advantage estimation enables the selection of reliable VLM-generated actions to guide policy learning, while mitigating the influence of unreliable ones.

The second challenge lies in avoiding unnecessary exploration once external knowledge is introduced, thereby achieving a better balance between exploitation and exploration. To address this, we design a VLM-guided Entropy Regularization (VER) module, which adjusts the entropy of the RL policy by integrating knowledge from both the policy itself and VLM-inferred actions. Since entropy controls the degree of exploration (Haarnoja et al., 2018a;b), VER reduces exploration when the RL policy aligns closely with reliable VLM-inferred actions – indicating that the learned policy is relatively effective – and increases exploration otherwise.

The contributions of our work are three-folds: (1) We analyze the issues of VLMs in guiding visual RL policy learning and provide insights into the key factors influencing the transfer of knowledge from large models to smaller models. (2) A VPG module in ERPV is proposed to estimate the reliability of VLMs at different states. This module dynamically selects reliable VLM-inferred actions to guide the policy learning process. (3) A VER module in ERPV is proposed to handle over-exploration by using the information related to visual RL policy and VLM-inferred actions, ensuring more efficient policy learning. Extensive experiments on multiple benchmarks (Carla (Dosovitskiy et al., 2017), DMControl (Tassa et al., 2018)) and CarRacing (Brockman, 2016) demonstrate that ERPV achieves superior performance. Experiment (Section 4.4) shows that **ERPV performs no worse than the base RL, even when the VLM is highly unreliable**, which proves the effectiveness of ERPV in adaptively identifying and utilizing reliable priors.

## 2    RELATED WORK

**LM-based Agents.**    The reasoning capabilities of LMs have spurred their adoption as core components in the development of autonomous agents across diverse domains (Zhen et al., 2023; Jeong et al., 2024; Biggie et al., 2023; Dasgupta et al., 2023; Kim et al., 2023). While LMs can generate decision plans from textual and visual inputs (Shinn et al., 2023; Pan et al., 2024; Zhang et al., 2024b; Fu et al., 2024a; Wang et al., 2024c), several challenges remain associated with their usage. These

challenges include the knowledge transduction gap, a scarcity of domain-specific knowledge, and computational overhead, among others. Current solutions include: (1) **Skill-grounded planning**, which integrated pre-trained skills with LMs to create executable plans (Mei et al., 2024; Brohan et al., 2023); (2) **Interactive fine-tuning**, which adapted environment-aware LMs through collected interaction data (Carta et al., 2023; Tan et al., 2024; Fu et al., 2024b); (3) **Efficiency optimization**, which employed RL-based input pruning (Nottingham et al., 2023) or query scheduling (Hu et al., 2023) to reduce the costs associated with invoking LMs. Our method differs from these approaches in three key aspects: (1) Previous methods require LMs during deployment, whereas our method distills knowledge from LMs into lightweight models; (2) Previous methods introduce additional action spaces during training, while our method avoids this issue; and (3) Our method optimizes based on LMs rather than relying entirely on them or discarding them. Notably, recent vision-and-language action (VLA) agents trained with RL have achieved significant progress and demonstrate strong performance (Hu et al., 2025; Guo et al., 2025; Chen et al., 2025c). However, they typically rely on large-scale expert data and require LMs at test time. In contrast, our method focuses on extracting partially reliable knowledge from general but imperfect VLMs and distilling it into compact RL policies.

**LM-Assisted RL.** Recent studies have explored the potential of leveraging general knowledge from LMs embeddings to support RL processes. These methods can be categorized into the following groups: (1) **Feature Extractors**. Pre-trained LMs encoded environmental states to enhance policy learning (Mezghani et al., 2023; Grigsby et al., 2023; Paischer et al., 2022). In contrast, we directly transfer the knowledge from LMs for policy guidance rather than for feature extraction. (2) **Policy Networks**. LMs were fine-tuned as RL policies through interactions with the environment (Li et al., 2022; Carta et al., 2023; Tan et al., 2024; Yao et al., 2023; Peiyuan et al., 2025; Shi et al., 2023), thereby achieving task alignment. In contrast to these methods, our framework eliminates the need for fine-tuning LMs, resulting in reduced computational costs. (3) **Hierarchical Planners**. LMs generated high-level task plans, while RL focuses on low-level control (Shukla et al., 2023; Chen et al., 2024; Xu et al., 2023; Du et al., 2023; Colas et al., 2023). For example, Dalal et al. (2024) employed LLMs for high-level planning, integrated motion planning for task sequencing, and utilized RL for low-level control learning. These methods may incur high deployment costs, whereas our method ultimately operates with a lightweight RL model. (4) **Reward Designers**. LMs created reward functions through image-text alignment (Huang et al., 2024), preference learning (Ghosh et al., 2025), or similar practices (Ma et al., 2023; Peng et al., 2025; Rocamonde et al., 2023; Wang et al., 2024b; 2025). However, the reward signals produced by LMs may be unstable, whereas our approach to policy knowledge transfer guarantees stable learning. (5) **Teaching Frameworks**. LMs teachers provided policy constraints (Xu et al., 2025a; Lee et al., 2025), curriculum demonstrations (Liu et al., 2024; Wu et al., 2024), or annealing guidance (Zhou et al., 2024). Our method differs by using adaptive knowledge filtering to address unreliable LMs inferences, thereby reducing error propagation and retaining valid guidance.

## 3 METHOD

### 3.1 PRELIMINARIES

The problem of visual RL is typically modeled as a Partially Observable Markov Decision Process $\mathcal{M}=< \mathcal{O}, \mathcal{S}, \mathcal{A}, \mathcal{P}, \mathcal{R}, \gamma >$, where $o_t \in \mathcal{O}$ defines visual observations, $s_t \in \mathcal{S}$ is the state feature which is extracted from $o_t$, $\mathcal{A}$ is the action space and $\mathcal{R}$ denotes the reward function. At each timestep $t$, the agent receives a visual observation $o_t$ and selects an action $a_t \in \mathcal{A}$ based on the policy $\pi(a_t|s_t)$. After executing the action, the agent receives a reward $r_{t+1} \sim \mathcal{R}(s_t, a_t)$, a next visual observation $o_{t+1}$ and a transition probability to the next state feature $s_{t+1} \sim \mathcal{P}(s_{t+1}|s_t, a_t)$. The objective is to learn an optimal policy $\pi^*$ that maximizes the expected cumulative return cumulative:

$$\mathcal{J}(\pi) = \sum_t \mathbb{E}_{s_t, a_t \sim \pi}[\gamma^t \mathcal{R}(s_t, a_t)], \tag{1}$$

where $\gamma \in [0, 1]$ is the discount factor. Our approach is based on Soft Actor-Critic (SAC) (Haarnoja et al., 2018a;b). During the process of maximizing cumulative rewards, SAC maintains an adaptive learning for the $\alpha$ parameter to control the entropy of policy, thereby enhancing the balance between exploration and exploitation. Based on this design, the objective optimal policy of SAC can be

formally defined as:

$$\pi^* = \arg\max_{\pi} \sum_t \mathbb{E}_{s_t, a_t \sim \pi} \left[ r(s_t, a_t) + \alpha \mathcal{H}(\pi(\cdot | s_t)) \right], \tag{2}$$

where $\alpha$ controls the trade-off between reward maximization and exploration, and $\mathcal{H}(\cdot)$ is the policy entropy. A larger value of $\alpha$ encourages higher entropy, promoting greater exploration, and vice versa. More detailed definitions can be found in the Appendix.

## 3.2 OVERVIEW

Our method leverages pre-trained VLMs to enhance sample efficiency in visual RL. During training, the VLM provides action suggestions based on the observation from training sample and a task-specific prompt, without interacting with the environment. These suggestions are used to guide policy learning and shape exploration dynamics. **At test time, the VLM is no longer needed, ensuring low inference latency for real-time deployment.**

At each timestep $t$, the visual RL agent interacts with the environment using its policy $\pi_\phi(\cdot)$, generating a transition $(o_t, a_{r,t}, r_t, o_{t+1}, d_t)$, where $a_{r,t}$ is the action sampled from the RL policy and $d_t$ is the termination mark. The observation $o_t$ and prompt are passed to the VLM to obtain an inferred action $a_{v,t}$, which is appended to the transition, forming an augmented trajectory element: $\tau_t = \{o_t, a_{r,t}, r_t, o_{t+1}, d_t, a_{v,t}\}$. $\tau_t$ are stored in the Replay buffer $\mathcal{B}$. During training, a batch of size $N$ are sampled from $\mathcal{B}$ for policy updates. To distinguish $t$, let $i$ represent the sample index in the batch, i.e., $i=1, 2, ..., N$.

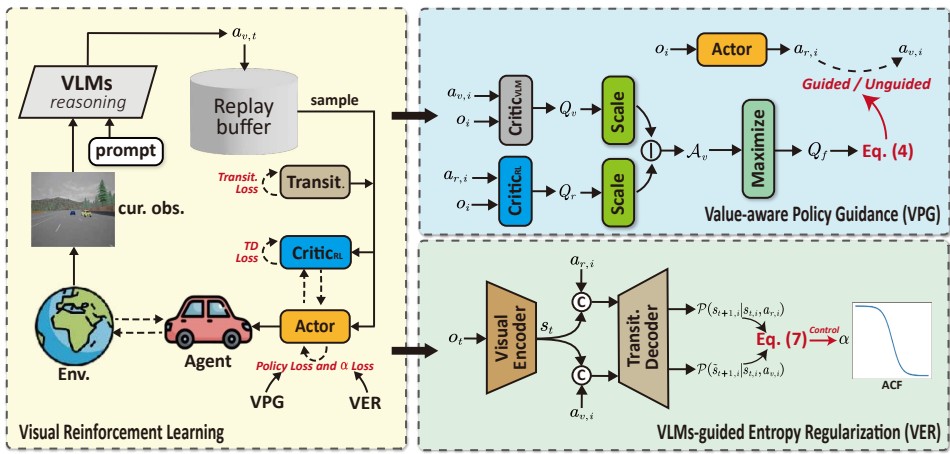

Figure 2: Framework of ERPV. During the training of visual RL, VLMs-inferred actions are integrated into the Replay buffer at time step $t$. The networks of **Actor**, **Critic$_{\mathbf{RL}}$**, and **Transition** are trained. In the VPG module, a fixed pre-trained network**Critic$_{\mathbf{VLM}}$** is introduced. By comparing the $Q$-value from both critic networks, we enhance the advantage estimation of the two types of actions during training. For the VER module, the networks of **Visual Encoder** and **Transition Decoder** are utilized to to derive a action confidence function for VLMs, which is then used as the regularization term for the adaptive learning of $\alpha$. At test time, the component of VLMs can be removed, leaving only the visual RL agent.

Fig. 2 shows the framework of ERPV. The framework includes four networks: the visual RL critic $\mathcal{C}_R(\cdot)$, the VLM critic $\mathcal{C}_V(\cdot)$, the actor $\pi_\phi(\cdot)$ and the transition model $\mathcal{T}(\cdot)$. These networks update their parameters by optimizing respective loss functions. To enhance visual RL using the partially reliable guidance provided by the VLM, two modules, including Value-aware Policy Guidance (VPG) and VLMs-guided Entropy Regularization (VER), are introduced to incorporate VLM knowledge into the actor loss. These modules improve learning without adding complexity at deployment. Experiments on different VLMs are detailed in Section 4.4, which proves that the performance of ERPV is not worse than that of base RL in the event of catastrophic failure cases. Detailed formulations of VPG and VER are provided in Section 3.3 and Section 3.4, respectively. Additional model specifications are given in the Appendix.

### 3.3 Value-aware Policy Guidance

The unreliability of VLM inferences could mislead the policy learning process in visual RL. To address this issue, we propose Value-aware Policy Guidance (VPG), which offers policy guidance selectively for certain states from a local perspective.

Firstly, to realize VPG, a critic network capable of evaluating the $Q$-value of VLMs-inferred actions is required. This pre-trained knowledge enables the critic network to provide more reliable $Q$-value estimates for VLMs, thereby accelerating the convergence process. Specifically, we interact with the environment using actions inferred by VLMs and train $\mathcal{C}_V(\cdot)$ with the collected data through temporal difference (Sutton, 1988) (TD) loss. It is important to note that $\mathcal{C}_V(\cdot)$ remains fixed during training to preserve its original evaluation capability, as a significant number of unstable trajectories from visual RL in the early stages can adversely affect its performance. Then, we design the VPG module based on the outputs of $\pi_\phi(\cdot)$, $\mathcal{C}_V(\cdot)$ and $\mathcal{C}_R(\cdot)$. The implementation of VPG consists of the following steps:

(1) $Q_r$ is obtained by inputting $a_{r,i}$ sampled from $\pi_\phi(\cdot)$ and $o_i$ into $\mathcal{C}_R(\cdot)$, i.e., $Q_r = \mathcal{C}_R(o_i, a_{r,i})$. (2) $Q_v$ is acquired by feeding $a_{v,i}$ and $o_i$ into $\mathcal{C}_V(\cdot)$, i.e., $Q_v = \mathcal{C}_V(o_i, a_{v,i})$. (3) $Q_r$ and $Q_v$ are processed through **Scale** Functions to ensure numerical stability and accelerate model convergence. We adopt a Min-Max Normalization (Patro & Sahu, 2015) as **Scale** to constrain the output values within the range $[0, 1]$. (4) The advantage of VLMs $\mathcal{A}_v$ is computed by $Q_r$ and $Q_v$, i.e., $\mathcal{A}_v = Q_v - Q_r$. $\mathcal{A}_v$ is then processed through a **Max** Function to produce a constraint factor $Q_f = \textbf{Max}(0, \mathcal{A}_v)$. (5) The constraint factor $Q_f$ is added to the policy constraint. The actor loss of VLMs guided visual RL could be formulated as:

$$\mathcal{L}_\mathcal{V} = Q_f \cdot \frac{1}{N} \sum_{i=1}^{N} (a_{v,i} - a_{r,i})^2. \tag{3}$$

Therefore, the final loss of $\pi_\phi(\cdot)$ training is integrated as:

$$\mathcal{L}_{policy} = \mathcal{L}_\pi + \lambda_1 \mathcal{L}_\mathcal{V}, \tag{4}$$

where $\lambda_1$ is the weight, and the loss $\mathcal{L}_\pi = \mathbb{E}_{a_t \sim \pi}[\alpha \log \pi(a_t|s_t) - Q(s_t, a_t)]$ is from the vanilla SAC (Haarnoja et al., 2018a). $\mathcal{L}_\mathcal{V}$ indicates that if $a_{v,i}$ is superior to $a_{r,i}$ in the current state, a certain level of supervision of VLMs is applied to constrain the visual RL policy; otherwise, it is ignored.

### 3.4 VLMs-guided Entropy Regularization

To reduce unnecessary exploration, we introduce VLM-guided Entropy Regularization (VER). Cross-validation between the VLM and RL policies enables adaptive exploration control: when the actions suggested by the VLM and the RL policy are highly consistent, they provide complementary evidence from both commonsense reasoning and value-based optimization, indicating high action reliability and justifying reduced exploration; otherwise, the opposite is true.

We design the Action Confidence Function (ACF, denoted $\mathcal{G}$) to guide entropy adjustment during training by assessing the consistency $\text{Dist}[a_{r,i}, a_{v,i}]$ between the RL policy action $a_{r,i}$ and the VLM-inferred action $a_{v,i}$. A lower $\mathcal{G}$ increases policy entropy to encourage exploration, while a higher $\mathcal{G}$ reduces entropy for exploitation. However, action values alone provide limited information about policy behavior. The state transition network $\mathcal{T}(\cdot)$ captures environmental dynamics by modeling $\mathcal{P}(s_{t+1}|s_t, a_t)$, offering richer representations than actions alone. Therefore, we use $\mathcal{T}(\cdot)$ to better assess the consistency between $a_{r,i}$ and $a_{v,i}$. Let $\mathcal{T}(\cdot) = \mathcal{P}(s_{t+1}|s_t, a_t)$ denote the learned transition dynamics. Thus, $\text{Dist}[a_{r,i}, a_{v,i}]$ can be transformed by the following formula:

$$\text{Dist}[a_{r,i}, a_{v,i}] \propto \text{Dist}\left[\mathcal{P}(s_{t+1,i}|s_{t,i}, a_{r,i}), \mathcal{P}(\tilde{s}_{t+1,i}|s_{t,i}, a_{v,i})\right], \tag{5}$$

where, $\tilde{s}_{t+1}$ is the predicted next state feature for $a_{v,i}$, and $\text{Dist}[\cdot]$ represents the distance between two elements. By incorporating state information, the robustness and accuracy of predictions are significantly enhanced.

Then, We define $\mathcal{G}$ based on the distance in predicted state transitions induced by $a_{r,i}$ and $a_{v,i}$. $\text{Dist}[\cdot]$ is calculated based on the L2 distance (Rüschendorf & Rachev, 1990). To be specific, $\mathcal{G}$ can be calculated by the following formula:

$$\mathcal{G} = \frac{1}{\|\mathcal{P}(s_{t+1,i}|s_{t,i}, a_{r,i}) - \mathcal{P}(\tilde{s}_{t+1,i}|s_{t,i}, a_{v,i})\|_2^2}, \tag{6}$$

where $\mathcal{G}$ indirectly measures the discrepancy between $a_{r,i}$ and $a_{v,i}$ through the transition network. $\mathcal{G}$ is defined to be inversely proportional to $\text{Dist}[\cdot]$: a smaller distance yields a higher confidence value, and vice versa.

Notably, $a_{r,i}$ interacts with the environment, yielding the true next state $s_{t+1,i}$. As training progresses, $\mathcal{T}(\cdot)$ becomes more accurate, and the VPG module encourages $a_{r,i}$ to align with the partially reliable $a_{v,i}$. This alignment allows the real transition $s_{t+1,i}$ from the Replay buffer to approximate $\mathcal{G} \approx \frac{1}{\|s_{t+1,i}-\mathcal{P}(\tilde{s}_{t+1,i}|s_{t,i},a_{v,i})\|_2^2}$, where $s_{t+1,i} \approx \mathcal{P}(s_{t+1,i}|s_{t,i},a_{r,i})$. The form of $\mathcal{G}$ follows the transition loss in DeepMDP (Gelada et al., 2019), detailed in the Appendix.

Given the distinct ranges of $\mathcal{G}$ and $\alpha$, a range transformation function is necessary. Since the Sigmoid function (Han & Moraga, 1995) can constrain values within the interval $(0, 1)$ and enables flexible control over the transition trend through parameter adjustment, we use the Sigmoid function (denoted as $\text{Sigmoid}(\cdot)$) to transform $\mathcal{G}$ into an appropriate target value for $\alpha$ (denoted as $\bar{\alpha}$), i.e., $\bar{\alpha}=\text{Sigmoid}(\mathcal{G})$. Thus, the loss of $\alpha$ training in ERPV can be modified as follows:

**Algorithm 1:** Relevant training pipeline

Initialize $\pi_\phi(\cdot), \mathcal{C}_R(\cdot), \mathcal{C}_V(\cdot), \mathcal{V}(\cdot), \mathcal{T}(\cdot)$, the VLMs and corresponding prompt, Replay buffer $\mathcal{B}$ and $\alpha$ as $\alpha_0$.
**for** Every episode **do**
    Reset environment and get initial observation $o_0$
    **for** Each time step $t$ **do**
        Execute VLMs to infer action $a_{v,t}$ based on the current observation image.
        Sample action $a_{r,t} \sim \pi_\phi(\cdot|o_t)$
        Execute action $a_{r,t}$, observe reward $r_t$, terminal state $d_t$ and next observation $o_{t+1}$
        Store transition $(o_t, a_{r,t}, r_t, o_{t+1}, d_t, a_{v,t})$ in $\mathcal{B}$
        **if** Enough samples in $\mathcal{B}$ **then**
            Sample $\{(o_{i,t}, a_{r,i}, r_i, o_{i,t+1}, d_i, a_{v,i})\}_{i=1}^N$
            Update $\mathcal{C}_R(\cdot)$ via TD loss
            Update $\mathcal{T}(\cdot)$ referred to DeepMdp (Gelada et al., 2019)
            Update $\pi_\phi(\cdot)$ via Equation (4)
            Update entropy coefficient $\alpha$ via Equation (7)
        **end if**
    **end for**
**end for**

$$\mathcal{L}_{\bar{\alpha}} = \mathcal{L}_\alpha + \lambda_2 \mathbb{E}_{(\alpha,\bar{\alpha}) \sim \mathcal{D}} \left[ (\alpha - \bar{\alpha})^2 \right], \qquad (7)$$

where the loss $\mathcal{L}_\alpha = \mathbb{E}_{a_t \sim \pi}[-\alpha \cdot (\log \pi(a_t|s_t) + \bar{\mathcal{H}})]$ is from the vanilla SAC Haarnoja et al. (2018a), $\bar{\mathcal{H}}$ is the target entropy, $\lambda_2$ is the weight, and $\mathcal{D}$ is the data distribution. By introducing a regularization term, RL can dynamically adjust the entropy based on ACF during training, thereby better balancing the trade-off between exploration. Algorithm 1 gives the relevant training pipeline.

## 4 EXPERIMENTS

### 4.1 EXPERIMENTAL SETTINGS

**Benchmarks.** To assess performance on diverse complex tasks, we used Carla (Dosovitskiy et al., 2017), DMControl in Mujoco (Tassa et al., 2018) and CarRacing in OpenAI Gym (Brockman, 2016) to assess model capabilities. As shown in Fig. 3, Carla simulate realistic driving scenarios, where VLMs are likely to provide reliable reasoning. In contrast, DMControl tasks involve high-dimensional synthetic environments that differ significantly from the real-world scenes in VLM pretraining data, resulting in weaker inference. By evalu-

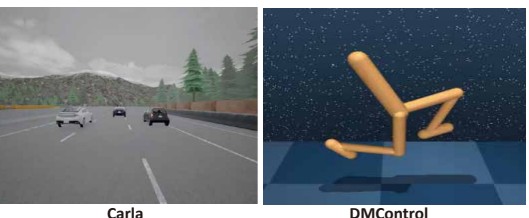

**Carla**          **DMControl**

Figure 3: Visualization of the Carla and DMControl benchmarks.

ating across diverse settings with varying VLM prior strength, we demonstrate that ERPV consistently improves performance. (1) **Carla** provides two high-dimensional visual scenarios: Highway (#HW) and Jaywalk (#JW). In the #HW scenario, the ego vehicle navigates around four randomly placed vehicles to maximize travel distance while avoiding collisions. In the #JW scenario, it must avoid three pedestrians who are crossing randomly in different locations. (2) For **DMControl**, we evaluate on Cheetah Run, Walker Walk (6D actions), Ball-in-Cup Catch, and Reacher Easy (2D actions)—tasks characterized by high-dimensional control, complex dynamics, and sparse rewards (Tang et al., 2024). The introduction to **CarRacing**, along with the additional benchmark details, can be found in the Appendix.

**Compared Methods.** For Carla, we conducted an extensive comparison of ERPV with a variety of methods, including vanilla visual RL, VLMs-based executor (VBE) (Mei et al., 2024), and

VLM-assisted visual RL. Among them, vanilla visual RL methods consist of: (i) a visual RL baseline SAC (Haarnoja et al., 2018b), (ii) a data augmentation-based visual RL method Drq (Kostrikov et al., 2020), (iii) Self-supervised learning for visual RL methods, including DeepMDP (Gelada et al., 2019), Curl (Laskin et al., 2020a), MLR (Yu et al., 2022), MaDI (Grooten et al., 2024) and ResAct (Liu et al., 2025). For VLM-assisted visual RL methods, we introduced previous the state-of-the-art (SOTA) baselines that extract knowledge from VLMs, including methods that supervise the agent directly through loss functions (DSF) (Lee et al., 2025), methods that distill VLMs-based decision knowledge into visual RL via annealing-based supervision functions (ASF) (Zhou et al., 2024), and a method that distill VLMs-guided representation into visual RL (DGC) (Xu et al., 2025b). To better highlight the advantages of ERPV, we also designed a method that provides a random supervision function (RSF), where $Q_f$ is randomly generated. For DMControl, we also compared the previous SOTA methods of vanilla visual RL, RSF and the VLMs-based methods to verify the effectiveness of our method.

**Implementation Details.** The proposed method is implemented based on the SAC algorithm (Haarnoja et al., 2018b). As for VLMs, Qwen2-VL (Wang et al., 2024a) is a widely adopted VLMs for inference tasks (Huo et al., 2025; Chen et al., 2025b; Guo et al., 2024). Thus, we employed Qwen2-VL-7B for prior knowledge reasoning tasks in the baseline experiments. The prompt in VLMs for each scenario is listed in the Appendix. $\lambda_1$ and $\lambda_2$ are set to 10.0 to balance the contribution of different loss terms, as suggested in prior

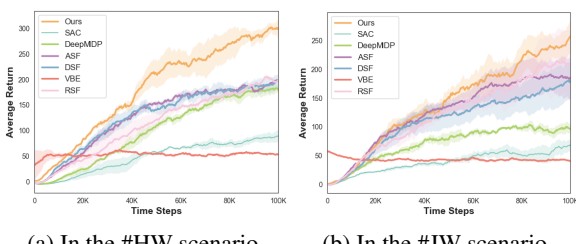

(a) In the #HW scenario.  (b) In the #JW scenario.

Figure 4: Training performance of some SOTA methods in the Carla benchmarks.

work (Xu et al., 2024b). Following the experimental setup of prior work (Young & Pugeault, 2024; Xu et al., 2024a), we trained the model using three different random seeds. Each benchmark was trained for 100K steps, and the metrics were reported based on 20 evaluation episodes with varying seeds. The Appendix provides more implementation details, including model details, training parameters, etc.

Table 1: Testing performance comparison with SOTA methods on the Carla driving benchmarks. The best results for each metric are bolded.

| Type | Methods | #HW | | #JW | |
|------|---------|-----|-----|-----|-----|
| | | ER ↑ | DD ↑ | ER ↑ | DD ↑ |
| Vanilla RL | SAC (Haarnoja et al., 2018b) | 86 ± 43 | 105 ± 50 | 38 ± 29 | 40 ± 30 |
| | DeepMDP (Gelada et al., 2019) | 167 ± 89 | 181 ± 93 | 82 ± 51 | 85 ± 52 |
| | Curl (Laskin et al., 2020a) | 118 ± 57 | 133 ± 62 | 78 ± 53 | 81 ± 53 |
| | Drq (Kostrikov et al., 2020) | 192 ± 51 | 205 ± 54 | 110 ± 68 | 113 ± 69 |
| | MLR (Yu et al., 2022) | 109 ± 51 | 132 ± 61 | 118 ± 50 | 121 ± 51 |
| | MaDI (Grooten et al., 2024) | 162 ± 62 | 172 ± 64 | 83 ± 47 | 86 ± 49 |
| | Dreamer-V3 (Hafner et al., 2025) | 182 ± 58 | 192 ± 62 | 141 ± 48 | 143 ± 48 |
| | ResAct (Liu et al., 2025) | 225 ± 48 | 238 ± 50 | 212 ± 54 | 216 ± 55 |
| Only VLMs | VBE (Mei et al., 2024) | 46 ± 36 | 29 ± 28 | 41 ± 19 | 42 ± 19 |
| VLM-assisted visual RL | DSF (Xu et al., 2024b) | 208 ± 82 | 220 ± 86 | 194 ± 60 | 195 ± 60 |
| | ASF (Zhou et al., 2024) | 207 ± 86 | 221 ± 90 | 206 ±52 | 208 ± 52 |
| | DGC (Xu et al., 2025b) | 199 ± 9 | 233 ± 16 | 146 ± 14 | 169 ± 18 |
| | RSF | 180 ± 91 | 189 ± 94 | 204 ± 77 | 205 ± 77 |
| | ERPV (Ours) | **299 ± 106** | **311 ± 109** | **268 ± 74** | **271 ± 75** |

## 4.2 RESULTS

(1) **Results on Carla**: Fig. 4 and Table 1 demonstrate the superiority of ERPV in Carla benchmarks across training and testing, evaluated through the mean and variance of episodic return (ER)

and driving distance (DD, in meters). Three key findings emerge from this analysis: (1) ERPV achieves state-of-the-art performance and improves the training efficiency by effectively integrating reliable VLMs knowledge in complex high-dimensional observation scenarios; (2) Improvements in baseline models (DSF/ASF) remain limited due to uncritical absorption of knowledge; and (3) The ablation study of the RSF confirms the statistical significance of the VPG module. (2) **Results on DMControl**: As shown in Table 2, ERPV also maintains performance advantages in DMControl tasks involving high-dimensional actions, complex motor control, and sparse rewards, effectively leveraging partially reliable knowledge from VLMs. Despite the VBE performed poorly overall, ERPV could nevertheless extract more reliable action priors from the VLM in specific states to enhance visual RL. Extended experimental results, including CarRacing, hyperparameters, hybrid VLM-RL trajectory, $\mathcal{C}_V(\cdot)$ and other related studies are all provided in the Appendix.

Table 2: Testing performance comparison with SOTA methods on the DMControl benchmark. The best results for each metric are are bolded.

| Methods / Metrics | | CartPole, Swing-up | Cheetah, Run | Walker, Walk | Ball in cup, Catch | Reacher, Easy |
|---|---|---|---|---|---|---|
| SAC (Haarnoja et al., 2018b) | ER ↑ | 237 ± 49 | 118 ± 13 | 95 ± 19 | 85 ± 130 | 239 ± 183 |
| DeepMDP (Gelada et al., 2019) | ER ↑ | 389 ± 44 | 306 ± 25 | 384 ± 197 | 704 ± 24 | 471 ± 173 |
| Curl (Laskin et al., 2020a) | ER ↑ | 582 ± 146 | 299 ± 48 | 403 ± 24 | 769 ± 43 | 538 ± 233 |
| Drq (Kostrikov et al., 2020) | ER ↑ | 759 ± 92 | 344 ± 67 | 612 ± 164 | 913±53 | 601 ± 213 |
| RAD (Laskin et al., 2020b) | ER ↑ | 694 ± 28 | 364 ± 38 | 552 ± 87 | 825 ± 49 | 734 ± 87 |
| MLR (Yu et al., 2022) | ER ↑ | 806 ± 48 | 482 ± 38 | 643 ± 114 | 933 ± 16 | 866 ± 103 |
| MaDI (Grooten et al., 2024) | ER ↑ | 704 ± 54 | 432 ± 44 | 574 ± 94 | 884 ± 36 | 766 ± 101 |
| ResAct (Liu et al., 2025) | ER ↑ | 819 ± 44 | 503 ± 42 | 772 ± 65 | 948 ± 44 | 917 ± 59 |
| Dreamer (Hafner et al., 2019) | ER ↑ | 781 ± 0 | 235±137 | 277±12 | 246±174 | 314±155 |
| VBE (Mei et al., 2024) | ER ↑ | 9 ± 10 | 5 ± 3 | 28 ± 13 | 56 ± 209 | 92 ± 131 |
| DSF (Xu et al., 2024b) | ER ↑ | 68 ± 14 | 257 ± 27 | 239 ± 65 | 659 ± 200 | 214 ± 347 |
| ASF (Zhou et al., 2024) | ER ↑ | 76 ± 6 | 347 ± 33 | 661 ± 39 | 857 ± 159 | 297 ± 426 |
| RSF | ER ↑ | 596 ± 16 | 278 ± 9 | 368 ± 35 | 507 ± 238 | 465 ± 305 |
| ERPV (Ours) | ER ↑ | **855 ± 2** | **516 ± 8** | **791 ± 26** | **953 ± 43** | **932 ± 66** |

Table 3: Comprehensive ablation experiments conducted in the #HW scenario validate the effectiveness of ERPV. The best results for each metric are bolded.

| Type | Idx. | ER ↑ | DD ↑ | Description |
|---|---|---|---|---|
| Effect of VPG module. | M1 | 174 ± 80 | 184 ± 81 | W/o VPG, i.e., removing $\mathcal{L}_\mathcal{V}$ |
| | M2 | 273 ± 136 | 285 ± 139 | W/o $Q_f$ in VPG |
| Effect of VER module. | M3 | 239 ± 73 | 248 ± 76 | W/o VER, i.e., removing $\mathbb{E}_{(\alpha,\bar{\alpha})\sim\mathcal{D}}\left[(\alpha - \bar{\alpha})^2\right]$ |
| | M4 | 249 ± 91 | 258 ± 93 | $\mathcal{G}$ is replaced as the MSE loss of $a_{v,i}$ and $a_{r,i}$ |
| | M5 | 167 ± 88 | 177 ± 90 | Change $a_{v,i}$ from $\mathcal{G}$ to actions sampled from $\pi_\phi(\cdot)$ |
| | M6 | 274 ± 103 | 286 ± 105 | Force $\alpha$ to 0 after 50k steps |
| | M7 | 241 ± 94 | 251 ± 96 | Only removing $\mathcal{L}_\alpha$ |
| | M8 | 247 ± 92 | 260 ± 95 | Use $\mathcal{P}(s_{t+1,i}|s_{t,i}, a_{r,i})$ to calculate $\mathcal{G}$ instead of $s_{t+1,i}$ |
| Effect of pre-training. | M9 | 174 ± 63 | 188 ± 67 | $\mathcal{C}_V(\cdot)$ is replaced by $\mathcal{C}_R(\cdot)$ |
| Ours | ERPV | **299 ± 106** | **311 ± 109** | Full version |

## 4.3 ABLATION STUDY

Table 3 validates the design of ERPV through ablation studies in the #HW scenario, utilizing the control variable method. The ablation studies can be divided into the following components: (1) **Effect of the VPG module**. A comparison between **M1/M2** and **M3**/DSF (see Table 1) confirms that the VPG module significantly enhances performance by suppressing unreliable VLMs-inferred actions while leveraging valid knowledge to guide visual RL policy learning. (2) **Effect of the VER module**. (i) **M3** directly demonstrates that the VER module enhances exploration efficiency; (ii) **M4** verifies that the action-state coupled ACF reduces prediction noise for improved decision accuracy; (iii) **M5** and **M6** jointly validate the VLMs-oriented ACF design. **M5** demonstrates that ACF necessitates the integration of prior knowledge from VLMs. **M6** indicates that ACF regularization facilitates adaptive entropy guidance, extending beyond mere entropy reduction; (iv) **M7** establishes the importance of native visual RL entropy control for maintaining a balance between exploration and exploitation; (v) **M8** verifies that using the actual $s_{t+1,i}$ to calculate $\mathcal{G}$ can enhance training stability and yield improved results. (3) **Effect of Pre-training**. **M9** demonstrates that $\mathcal{C}V(\cdot)$ and

$\mathcal{CR}(\cdot)$ play distinct roles, with the former dedicated to evaluating the VLM policy and the latter to evaluating the RL policy. The ablation experiment confirmed the effectiveness and rationale of the designed ERPV from all perspectives.

### 4.4 VLMs Scale and Worst Cases Study

We evaluate ERPV using VLMs of varying scales: Qwen2-VL-7B, Qwen2-VL-2B (Wang et al., 2024a), and LLava-1.5-7B (Liu et al., 2023), with Qwen2-VL-2B used to assess performance under weaker VLM priors. To assess robustness in catastrophic failure scenarios, we introduce a Random Executor (random policy) and a Worst Model, which sets all output values of the VLM to 0, causing both ER and DD to be 0 in every state. Results in Table 4 show that: (1) VBE performance follows Qwen2-VL-7B > LLava-1.5-7B > Qwen2-VL-2B, indicating that ERPV benefits from stronger VLM scene understanding; (2) even when the Qwen2-VL-2B underperform the random executor, ERPV

Table 4: Testing performance of VLMs with different model sizes in the #HW scenario. The best results for each metric are bolded.

| Type | Models | ER ↑ | DD ↑ |
|---|---|---|---|
| Random | Random Executor | 2 ± 9 | 26 ± 17 |
| Based RL | DeepMDP | 167 ± 89 | 181 ± 93 |
| VBE | Worst Model | 0 ± 0 | 0 ± 0 |
| | LLava-1.5-7B | 22 ± 30 | 29 ± 28 |
| | Qwen2-VL-2B | 1 ± 5 | 2 ± 5 |
| | Qwen2-VL-7B | 46 ± 36 | 47 ± 37 |
| ERPV (Ours) | Worst Model | 170 ± 69 | 184 ± 72 |
| | LLava-1.5-7B | 203 ± 93 | 219 ± 96 |
| | Qwen2-VL-2B | 179 ± 97 | 194 ± 101 |
| | Qwen2-VL-7B | **299 ± 106** | **311 ± 109** |

achieves performance on par with the base RL. This robustness arises because, although the VLM may perform poorly overall, it can still provide reliable guidance in certain states; (3) even if the Worst Model is the worst, ERPV still maintains performance comparable to that of base RL. This highlights the advantage of ERPV's adaptive selection of prior guidance.

### 4.5 Visual analysis

We have provided two visual analyses to facilitate an intuitive understanding of the functional processes within the VPG and VER modules. Fig. 5 (a) shows the evolution of the average $Q_f$. As training progresses, the RL policy gradually aligns with reliable VLM actions in certain states under VPG, reducing action discrepancy. Consequently, $Q_f$ decreases and converges to zero. Fig. 5 (b) shows the entropy coefficient $\alpha$ in VER. Compared to SAC, DeepMDP, and DSF, ERPV

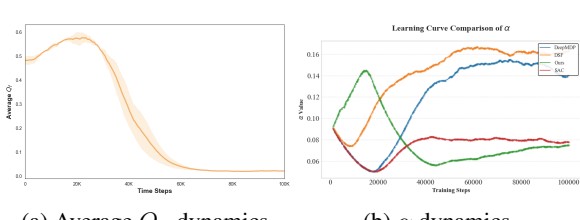

(a) Average $Q_f$ dynamics.    (b) $\alpha$ dynamics.

Figure 5: Illustration of Average $Q_f$ and $\alpha$ dynamics during #HW scenario training in CARLA.

yields two key observations: (1) it maintains higher $\alpha$ early in training, enabling rapid alignment with partially reliable VLM actions through enhanced exploration; (2) as the two actions approach each other in a specific state, $\alpha$ steadily decreases, reflecting a shift toward exploitation of verified actions. In contrast, baseline methods do not adapt $\alpha$ based on action reliability. The trend in $\alpha$ demonstrates VER's effective control mechanism, improving visual RL sample efficiency.

## 5 Conclusion

This paper investigates how to utilize the action commonsense reasoning results of partially reliable VLMs as supervision signals to guide the policy learning of visual RL. In this work, we present ERPV, a principled framework for enhancing visual RL by leveraging partially reliable VLM priors. Through value-aware policy guidance and dynamics-informed entropy regulation, our method effectively integrates external knowledge while improving the balance between exploration and exploitation, achieving superior performance across diverse complex vision-to-control tasks. Notably, ERPV demonstrates stable learning even under weak or catastrophically flawed VLMs' reasoning, highlighting its resilience to imperfect supervision. This work highlights the promise of knowledge-augmented RL and provides a foundation for future research on RL with imperfect supervision.

ETHICS STATEMENT

All the experiments in this work were conducted in a publicly available reinforcement learning simulation environment and did not involve private or sensitive information. The suggested approach is purely for academic research. Any deployment should carefully consider potential ethical risks, such as bias or abuse.

REPRODUCIBILITY STATEMENT

This work provides relevant codes in the supplementary materials, including the configuration of the running environment. The reinforcement learning environment used is also public, providing the possibility for reproducibility.

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

APPENDIX

The contents of this Appendix are presented as follows:

Section A explains why the decision reasoning of VLM is partially reliable and provides examples to illustrate.

Section B provides a more definitions of the proposed method, including SAC (Haarnoja et al., 2018a;b) and DeepMdp (Gelada et al., 2019), aiding readers in better understanding the foundation proposed in this paper.

Section C offers a detailed description of the experimental settings, including the reward configurations for each benchmark, the structure of the model network, the usage of VLMs, experimental parameters, and prompt details, among other aspects.

Section D analyzes the training method of $\mathcal{C}_V(\cdot)$.

Section E presents additional experimental findings (such as the experiments on CarRacing, hyperparameters, hybrid VLM-RL trajectory and hardware consumption) and offers pertinent conclusions and discussions.

Section F analyzes the proposed VPG and VER modules and presents the visual analysis.

Section G discusses the limitations and future works.

## A   DESCRIPTION OF THE VLMS' PARTIAL RELIABILITY

Vision-language models (VLMs) combine visual encoders with large language models (LLMs), enabling cross-modal understanding and semantic reasoning over image-based inputs. While it is unclear whether such models were explicitly trained on spatio-temporal action data, they are exposed to vast amounts of diverse visual-textual data during pretraining, endowing them with strong zero-shot generalization capabilities. Since action or decision making is a natural form of semantic reasoning, VLMs inherently support image-to-action mapping—even without fine-tuning. Prior works (Chen et al., 2025a; DeFazio et al., 2024; Zhao et al., 2024; Xu et al., 2024b; Aissi et al., 2025) have demonstrated this potential: (Chen et al., 2025a) use VLMs to infer control actions for robotic dogs from images, while (Xu et al., 2024b) leverage them for autonomous driving decisions. Like these efforts, our work aims to harness VLMs to guide agents in generating image-conditioned actions—either directly or through knowledge distillation into RL policies.

Our experiments primarily use Qwen2-VL-7B, a VLM trained on datasets including DocVQA, InfoVQA, RealWorldQA, and MTVQA (Wang et al., 2024a), with RealWorldQA containing driving-related QA pairs. However, CARLA lacks large-scale QA data, so the model is not specifically trained on this simulator. Nevertheless, the VLM's strong multimodal understanding enables it to extract relevant knowledge from analogous driving scenarios. As shown in Fig. 6 (a), it reliably infers "move forward" actions in straight-road, obstacle-free scenes, but fails in pedestrian avoidance due to inaccurate prediction of human motion, leading to collisions. In DMControl tasks—where pretraining data is even scarcer—the overall reliability of VLM actions is lower. Fig. 6 (b) illustrates that suggestions are only reliable in certain states, particularly in high-dimensional action-space tasks like Walker, Walk and Cheetah, Run. Crucially, our results show that ERPV effectively leverages these partially reliable priors, enhancing visual RL performance despite imperfect guidance.

In our work, VLMs may not be a good source of supervision (teacher). That is to say, it could be partially reliable or even very unreliable. Although VLMs are not entirely reliable, conventional RL start with completely random strategies, so it is helpful to draw inspiration from VLMs. Although VLMs may be not trained with a large number of image-action pairs related to Carla, due to the powerful generalization reasoning ability of VLMs, it can also infer better prior actions than random actions in certain states. Our approach combines the powerful generalization reasoning ability of VLMs with the later self-exploration and development of RL. When VLM infers better actions than the exploration actions of RL in certain states, the learning efficiency of RL policies can be accelerated through guidance. If there is expert data, VLM is indeed not necessary. However, expert data is extremely expensive, and VLMs offer a solution for handling such cases.

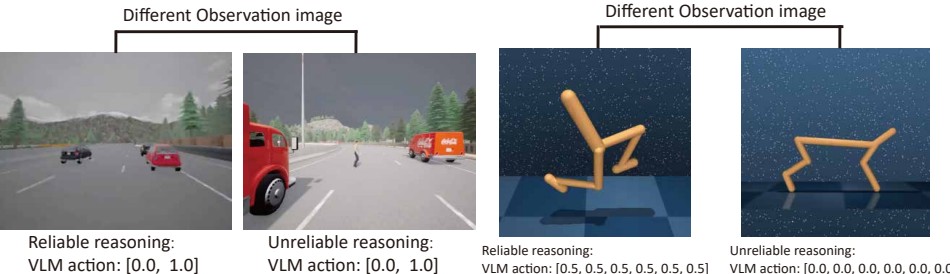

(a) In the autonomous driving.     (b) In the robot operation.

Figure 6: Examples of VLM-inferred actions for different tasks.

## B  MORE DEFINITIONS OF ERPV

The SAC algorithm (Haarnoja et al., 2018a;b) is a widely-used baseline method in RL, whose core idea is to optimize the policy to maximize cumulative rewards while incorporating a regularization term for policy entropy, thereby effectively balancing the trade-off between exploration and exploitation.

For training the agent, SAC repeatedly applies a modified Bellman backup operator $\mathcal{T}^\pi Q(s_t, a_t)$ to iteratively optimize the action-value function $Q$, as follows:

$$\mathcal{T}^\pi Q(s_t, a_t) \triangleq \mathbb{E}_{s_t, a_t \sim \pi}[Q(s_t, a_t) - (r_t + \gamma V(s_{t+1}))], \tag{8}$$

where

$$V(s_{t+1}) = \mathbb{E}_{a_{t+1} \sim \pi}[\tilde{Q}(s_{t+1}, a_{t+1}) - \alpha \log \pi(a_{t+1}|s_{t+1})]. \tag{9}$$

In this context, $\tilde{Q}(s_{t+1}, a_{t+1})$ denotes the target state-action value function, which is generally derived by applying an exponential moving average to the parameters of the function $Q$ to enhance training stability. In SAC, the critic network is updated using temporal difference (Sutton, 1988) (TD) loss and the actor network is optimized by minimizing the divergence between $Q$ and the policy itself:

$$\mathcal{L}_\pi = \mathbb{E}_{a_t \sim \pi}[\alpha \log \pi(a_t|s_t) - Q(s_t, a_t)]. \tag{10}$$

The update loss of parameter $\alpha$ is:

$$\mathcal{L}_\alpha = \mathbb{E}_{a_t \sim \pi}[-\alpha \cdot (\log \pi(a_t|s_t) + \bar{\mathcal{H}})], \tag{11}$$

where, $\bar{\mathcal{H}}$ is the target entropy, which is usually set to a negative value of the action space dimension.

In this work, we adopt the design of the Transition network from DeepMDP (Gelada et al., 2019) for predicting feature distribution of the next state. During visual RL training, a visual encoder (denoted as $\mathcal{V}(\cdot)$) is maintained to extract the feature $s_t$ from the input sampled observation $o_t$, i.e., $s_t = \mathcal{V}(o_t)$. A Transition network (denoted as $\mathcal{T}(\cdot)$) consists of $\mathcal{V}(\cdot)$ and a Decoder (denoted as $\mathcal{T}_D(\cdot)$). For predicting the next state feature, $o_{i,t}$ and $o_{i,t+1}$ are firstly input into $\mathcal{V}(\cdot)$. Then, $\mathcal{V}(\cdot)$ output $s_{i,t}$ and $s_{i,t+1}$, respectively. Next, the concatenated feature $[a_i, s_{i,t}]$ is input to $\mathcal{T}_D(\cdot)$. Finally, the predicted feature $\hat{s}_{i,t+1}$ output by $\mathcal{T}_D(\cdot)$ is compared with the ground-truth feature $s_{i,t+1}$ to compute the transition loss. Here, the feature $\hat{s}_{i,t+1}$ consists of two predicted Gaussian distribution features: mean ($\mu$) and variance ($\sigma$), i.e., $\hat{s}_{i,t+1} = \{\hat{\mu}_{i,t+1}, \hat{\sigma}_{i,t+1}\}$. Therefore, the transition loss can be formulated into:

$$\mathcal{L}_\mathcal{T}(s_{i,t+1}, \hat{\mu}_{i,t+1}, \hat{\sigma}_{i,t+1}) = \mathbb{E}_{i \in N}\Big[\Big(\frac{\hat{\mu}_{i,t+1} - s_{i,t+1}}{\hat{\sigma}_{i,t+1}}\Big)^2 + \log(\hat{\sigma}_{i,t+1})\Big], \tag{12}$$

where,

$$\begin{aligned} s_{i,t} &= \mathcal{V}(o_{i,t}) \\ s_{i,t+1} &= \mathcal{V}(o_{i,t+1}) \\ \hat{s}_{i,t+1} &= \{\hat{\mu}_{i,t+1}, \hat{\sigma}_{i,t+1}\} = \mathcal{T}_D([a_i, s_{i,t}]). \end{aligned} \tag{13}$$

When calculating Equation (6) in Section 3.4, we continue to utilize the loss calculation from the transition network in DeepMdp to compute $\mathcal{G}$. Let the output of $\mathcal{T}(o_{t,i}, a_{v,i})$ is $\mathcal{P}(\tilde{s}_{t+1,i}|s_{t,i}, a_{v,i}) = \{\tilde{\mu}_{i,t+1}, \tilde{\sigma}_{i,t+1}\}$. Then we can obtain the following formula:

$$\mathcal{G} \approx \frac{1}{\|s_{t+1,i} - \mathcal{P}(\tilde{s}_{t+1,i}|s_{t,i}, a_{v,i})\|_2^2} = \frac{1}{\mathcal{L}_\mathcal{T}(s_{i,t+1}, \tilde{\mu}_{i,t+1}, \tilde{\sigma}_{i,t+1})} \tag{14}$$

## C   DETAILED EXPERIMENTAL SETUP

### C.1   CARLA BENCHMARKS

In the visual RL framework, the agent's input consists of RGB visual observations. To ensure a fair comparison across experiments, we standardized the camera parameter configurations for all trials in the Carla simulation environment, with detailed settings provided in Table 5. Furthermore, to minimize the impact of environmental variables on the experimental results, all experiments were conducted under identical weather condition, i.e., Cloudy Noon. The max episode steps of these benchmarks is 1000.

Table 5: Some key settings of the RGB camera in the Carla benchmarks.

| Attributes | Value | Description |
|---|---|---|
| image size | [128, 128] | Width and height of the image in pixels. |
| fov | 60 | Horizontal field of view (FOV) of the camera. |
| tick | 20 | The RGB camera's capture frequency in hertz. |
| gamma | 2.2 | The gamma correction applied to the RGB camera's output. |
| iso | 100 | The camera sensor sensitivity. |
| exposure_min_bright | 10 | Minimum brightness for auto exposure. |
| exposure_max_bright | 12 | Maximum brightness for auto exposure. |
| motion_blur_intensity | 0.45 | Strength of motion blur. |

Our objective is to maximize the agent's travel distance within 100K steps while avoiding collisions. To achieve this, we designed two reward functions for #HW and #JW scenarios as follows:

(1) The reward function for #HW scenario is defined as follows:

$$r_t = \lambda_1 \cdot v_s \cdot d_t - \lambda_2 \cdot collision - \lambda_3 \cdot |steer|, \tag{15}$$

where, $v_s$ is the agent's velocity, $d_t$ is the simulation time difference (in second), and $collision$ represents the intensity of the collision, calculated by Carla's collision sensor. The coefficients $\lambda_1 = 1$, $\lambda_2 = 10^{-4}$, and $\lambda_3 = 1$ are used to balance these components. The first term of the reward function encourages the agent to stay within the lane and maximize travel distance. The second and third terms penalize collisions and over-steering, respectively, guiding the agent to learn safer and more stable driving policies.

(2) The reward function for #JW scenario is defined as follows:

$$r_t = \lambda_1 \cdot v_s \cdot d_t - \lambda_2 \cdot collision - \lambda_3 \cdot lane\_invasion - \lambda_4 \cdot |steer|, \tag{16}$$

where, $v_s$ is the agent's velocity, $d_t$ is the simulation time difference (in second), $collision$ represents the intensity of the collision, calculated by Carla's collision sensor, and $lane\_invasion$ represents the intensity of lane invasion, which is captured in real-time by the built-in lane detection sensor in the Carla simulation platform. The coefficients $\lambda_1 = 1$, $\lambda_2 = 10^{-3}$, $\lambda_3 = 10^{-2}$, and $\lambda_4 = 0.1$ are used to balance these components. It is worth noting that the third item provides a quantitative measure of the agent's lane-keeping ability.

The action of the agent is $[steer, throttle]$ in Carla (Dosovitskiy et al., 2017). The steering control ($steer$) ranges from -1 to 1, where values less than 0 indicate a left turn, values equal to 0 indicates going straight and values greater than 0 indicate a right turn. Similarly, the throttle control ($throttle$) ranges from -1 to 1, with values $\leq 0$ representing deceleration or braking, and values greater than 0 corresponding to varying levels of acceleration.

## C.2 OTHER BENCHMARKS

In both the DMControl (Tassa et al., 2018) and CarRacing environments, the input image size is standardized to [84, 84], with RGB configurations following the default settings of the gym framework, and the reward function is built into the gym library (Brockman, 2016). This setup ensures consistency and reproducibility across experiments, while aligning with standard practices in mainstream reinforcement learning research. Notably, the CarRacing experiments are conducted using the CarRacing-V1 version. The max episode steps of these benchmarks is 1000.

In the DMControl tasks (Tassa et al., 2018), we evaluate Cheetah Run, Walker Walk (both with 6D continuous actions), Ball-in-Cup Catch (with 2D continuous actions), and Reacher Easy (with 2D continuous actions), which feature high-dimensional actions, complex motor control, or sparse rewards (Tang et al., 2024), offering diverse distributions beyond autonomous driving. Following the RAD setup (Laskin et al., 2020b), we train for 100k steps with three different seeds and report the mean and variance over 20 evaluation runs.

In the CarRacing environment (Brockman, 2016), the action space is continuous and consists of three dimensions: steering, acceleration, and braking. The steering value ranges from $[-1, 1]$, where -1 corresponds to a full left turn and 1 represents a full right turn. The acceleration and braking values are also within the range [0, 1], with acceleration controlling the throttle and braking controlling the brake intensity. This continuous action space allows for fine-grained control of the car's movement.

## C.3 MODEL DETAILS

Our models are built upon the DeepMDP framework (Gelada et al., 2019), with three core networks being critical to our method: a Critic network, a Actor network, and a Transition network. The architecture and input-output design of these models are as follows: (1) The critic network ($\mathcal{C}_R(\cdot)$ or $\mathcal{C}_V(\cdot)$) consists of a Visual Encoder $\mathcal{V}(\cdot)$ and Q Networks, taking the observation and action as inputs and outputting the $Q$-value. We employ the SAC algorithm, where the critic network adopts a dual-Q structure, and the final $Q$-value is the minimum of the outputs from the two Q-networks. (2) The actor network $\pi_\phi(\cdot)$, also composed of a Visual Encoder $\mathcal{V}(\cdot)$ and a Action Network, takes the observation as input and outputs an action sampled from the policy distribution. (3) The transition network $\mathcal{T}(\cdot)$, comprising a Visual Encoder $\mathcal{V}(\cdot)$ and Decoder, takes the observation and action as inputs and outputs the predicted next state feature. Fig. 7 illustrates the workflow of $\mathcal{C}_R(\cdot)$ (or $\mathcal{C}_V(\cdot)$), $\pi_\phi(\cdot)$, and $\mathcal{T}(\cdot)$. Note that the above visual encoders share parameters during visual RL training. In addition, we freeze the transition network (including $\mathcal{V}(\cdot)$ and $\mathcal{T}_D(\cdot)$) when calculating the output of the VER.

We utilize the torchinfo library in PyTorch to display detailed information of each network. The results, shown in Fig. 8, Fig. 9, Fig. 10, Fig. 11, include the network architecture, output shapes, and the number of parameters.

## C.4 THE USAGE SETTINGS OF VLMS

In our method, the reasoning actions of VLMs do not engage in environmental interactions; instead, they offer supplementary guidance, thereby not influencing the interaction speed between visual RL and the environment. This guiding characteristic allows VLMs to be utilized in both real-world and simulated environments. There are two approaches to invoking VLMs: (1) Collect a substantial amount of Replay buffer and input the observed images sampled from the Replay buffer and prompts into VLMs to generate reasoning results at a specified frequency. (2) Conduct step-by-step reasoning at a specified frequency during the interaction process when the visual RL agent interact with the environment before adding data to the Replay buffer. Since these two approaches are nearly equivalent, they do not significantly impact the final performance. In the experiment, Option Two was selected, with the frequency set to 10, meaning the VLMs was invoked once every 10 frames.

## C.5 HYPERPARAMETERS SETTINGS

Our research intentionally avoids the use of any image augmentation algorithms to unilaterally validate the enhancement capability of VLMs on the performance of visual RL policies. The state feature $s_t$ is obtained by stacking 3 consecutive frames of observed images and passing them through

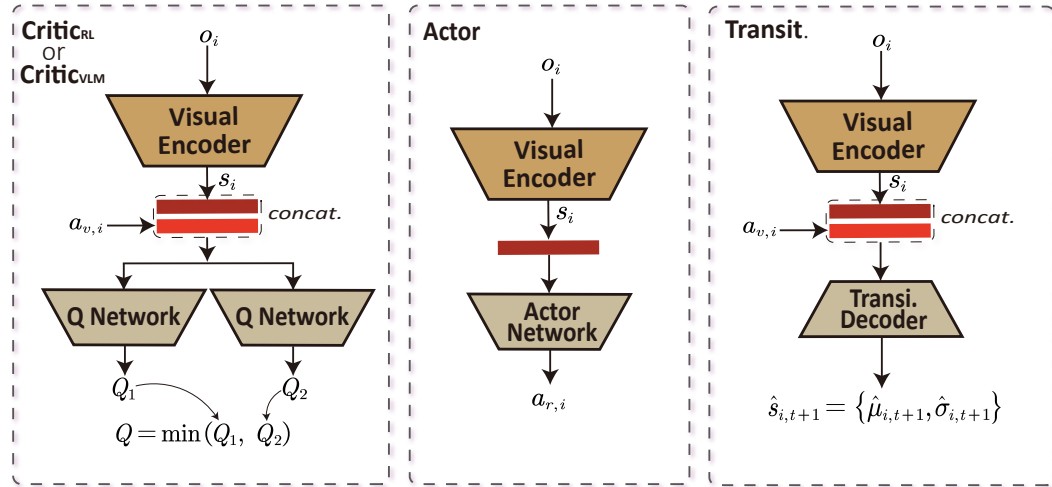

Figure 7: The workflow of $\mathcal{C}_R(\cdot)$ (or $\mathcal{C}_V(\cdot)$), $\pi_\phi(\cdot)$, and $\mathcal{T}(\cdot)$. All visual encoders share parameters during visual RL training.

```
====================================================================================
Layer (type:depth-idx)          Output Shape        Param #
====================================================================================
PixelEncoderCarla               [1, 50]             --
├─ModuleList: 1-1               --                  --
│    └─Conv2d: 2-1              [1, 64, 62, 62]      14,464
│    └─Conv2d: 2-2              [1, 128, 30, 30]     73,856
│    └─Conv2d: 2-3              [1, 256, 14, 14]     295,168
│    └─Conv2d: 2-4              [1, 256, 6, 6]       590,080
├─Linear: 1-2                   [1, 50]             460,850
├─LayerNorm: 1-3                [1, 50]             100
====================================================================================
Total params: 1,434,518
Trainable params: 1,434,518
Non-trainable params: 0
Total mult-adds (M): 201.63
====================================================================================
Input size (MB): 0.59
Forward/backward pass size (MB): 3.37
Params size (MB): 5.74
Estimated Total Size (MB): 9.69
====================================================================================
```

Figure 8: The structure, output shape, and number of parameters for Visual Encoder $\mathcal{V}(\cdot)$.

$\mathcal{V}(\cdot)$, i.e., $s_t = \{o_t, o_{t-1}, o_{t-2}\}$. Table 6 provides some other key hyperparameters in the Carla experiments. The parameter configuration and image enhancement methods of DMControl are consistent with those of RAD (Laskin et al., 2020b). In the CarRacing experiments, the main difference is that RGB frame dimensions are changed to $84{\times}84{\times}3$, and the learning rate of critic and $\alpha$ is changed to $20^{-4}$. Notably, in the warm-up phase, the agent could explore using the VLMs-inferred actions because it may be superior to the initial random policy.

## C.6 PROMPTS DETAILS

The benchmarks for each scenario of Carla are shown in the Fig. 12 and Fig. 13. In the Carla scenario's prompt, we dynamically compute the speed and yaw of the ego vehicle in real-time and incorporate them as part of the vehicle's state description within the prompt. Fig. 14 illustrates an example of VLMs reasoning in the #HW scenario. The VLMs implementation loads the prompt designed in Fig. 12, where the user simply enters a picture of the current observation. We first employ VLMs to describe the image content to facilitate action inference, and then extract throttle and steering information from their responses.

```
================================================================================
Layer (type:depth-idx)          Output Shape        Param #
================================================================================
Sequential                      [1, 4]              --
├─Linear: 1-1                   [1, 1024]           52,224
├─ReLU: 1-2                     [1, 1024]           --
├─Linear: 1-3                   [1, 1024]           1,049,600
├─ReLU: 1-4                     [1, 1024]           --
├─Linear: 1-5                   [1, 4]              4,100
================================================================================
Total params: 1,105,924
Trainable params: 1,105,924
Non-trainable params: 0
Total mult-adds (M): 1.11
================================================================================
Input size (MB): 0.00
Forward/backward pass size (MB): 0.02
Params size (MB): 4.42
Estimated Total Size (MB): 4.44
================================================================================
```

Figure 9: The structure, output shape, and number of parameters for the Decoder of $\pi_\phi(\cdot)$.

```
================================================================================
Layer (type:depth-idx)          Output Shape        Param #
================================================================================
Sequential                      [1, 1]              --
├─Linear: 1-1                   [1, 1024]           54,272
├─ReLU: 1-2                     [1, 1024]           --
├─Linear: 1-3                   [1, 1024]           1,049,600
├─ReLU: 1-4                     [1, 1024]           --
├─Linear: 1-5                   [1, 1]              1,025
================================================================================
Total params: 1,104,897
Trainable params: 1,104,897
Non-trainable params: 0
Total mult-adds (M): 1.10
================================================================================
Input size (MB): 0.00
Forward/backward pass size (MB): 0.02
Params size (MB): 4.42
Estimated Total Size (MB): 4.44
================================================================================
```

Figure 10: The structure, output shape, and number of parameters for the Decoder of $\mathcal{C}_R(\cdot)$ (or $\mathcal{C}_V(\cdot)$).

```
================================================================================
Layer (type:depth-idx)          Output Shape        Param #
================================================================================
DeterministicTransitionModel    [1, 50]             --
├─Linear: 1-1                   [1, 512]            27,136
├─LayerNorm: 1-2                [1, 512]            1,024
├─Linear: 1-3                   [1, 50]             25,650
================================================================================
Total params: 53,810
Trainable params: 53,810
Non-trainable params: 0
Total mult-adds (M): 0.05
================================================================================
Input size (MB): 0.00
Forward/backward pass size (MB): 0.01
Params size (MB): 0.22
Estimated Total Size (MB): 0.22
================================================================================
```

Figure 11: The structure, output shape, and number of parameters for the Decoder of $\mathcal{T}(\cdot)$.

Table 6: Some key hyperparameters used in the Carla benchmarks.

| Hyperparameter | Value |
|---|---|
| RGB frame dimensions | $128 \times 128 \times 3$ |
| Action repeat | 4 |
| Frame stack | 3 |
| Initial sampling steps (warming up) | 1,000 |
| Total training steps | 100,000 |
| Evaluation episodes | 20 |
| Replay buffer size | 100,000 |
| Initial $\alpha$ ($\alpha_0$) | 0.1 |
| Learning rate of $\alpha$ | $10^{-4}$ |
| Learning rate of Actor | $10^{-3}$ |
| Learning rate of Critic | $10^{-3}$ |
| Learning rate of Visual Encoder | $10^{-3}$ |
| Learning rate of transition network | $10^{-3}$ |
| Optimizer for Actor, Critic and $\alpha$ | Adam (betas=(0.9, 0.999)) |
| Optimizer for Visual encoder and Auxiliary tasks | Adam |
| Batch size | 128 |
| VLMs-guided supervision update frequency | 10 |
| Transition and reward auxiliary task update frequency | 1 |
| Actor update frequency | 2 |
| Critic target update frequency | 2 |

Figure 12: The prompt of VLMs in the #HW scenario.

**System Message:**

You are an AI assistant integrated into an autonomous driving system in Carla. Your task is to analyze the RGB camera image taken from the front of the ego vehicle and output throttle and steering values.

**Prompt:**

The scene where ego is located is a high-speed scene in automatic driving. Ego needs to avoid collisions on highways, try to stay in the same lane. Please describe the image in detail to determine whether there is a mmediate danger of collision. If so, ego needs to consider slowing down (set throttle <0). Furthermore, please describe the image in detail to determine if the ego is going through a bend. If so, the ego's steering needs to be adjusted. It should be noted that the throttle range is -1 ~ 1, the throttle <=0 indicates slowing down, and the throttle >0 indicates different acceleration. Steering ranges from-1 to 1, with steering >0 indicating a right turn, steering <0 indicating a left turn, and steering=0 indicating straight driving. In addition to the image, ego's current speed is {:.2f}m/s, yaw is {:.2f}. If there is no immediate danger of collision, ego can set throttle=1 to accelerate. Finally, design reasonable throttle and steering output values for the ego based on the description of the image, the current speed and yaw. Please follow the following template to output content:\nDescribe:xxx \nthrottle:xxx \nsteer:xxx

## D    SPECIFIC ANALYSIS OF $\mathcal{C}_V(\cdot)$

### D.1    THE PRETRAINED METHOD FOR $\mathcal{C}_V(\cdot)$

As mentioned in Method Section, $\mathcal{C}_V(\cdot)$ plays a crucial role. In this section, we provide a detailed description of the pre-training method for $\mathcal{C}_V(\cdot)$ employed to enhance the performance of ERPV.

Existing approaches for training critic networks to evaluate VLMs-inferred actions complex scenes face a critical challenge: incomplete state-action exploration. Sole reliance on VLMs-inferred actions often restricts the agent to limited scenarios (e.g., straight lanes or low-speed maneuvers), resulting in sparse state-space coverage and suboptimal critic generalization. To address this, we propose a Random Alternating Exploration (RAE) method that hybridizes a pretrained visual RL (e.g., DeepMDP) actor and VLMs actor. Key insights of RAE include:

**System Message:**

You are an AI assistant integrated into an autonomous driving system in Carla. Your task is to analyze the RGB camera image taken from the front of the ego vehicle and output throttle and steering values.

**Prompt:**

The scene where ego vehicles are located is the ghost probe scene, that is, pedestrians suddenly cross the road. Please describe the content of the image, determine if there is a pedestrian likely to appear, or cross the road, if there is, and the distance is close, ego need to consider adjusting action to avoid collision. It should be noted that the throttle range is [-1, 1], the throttle <=0 indicates slowing down, and the throttle >0 indicates different acceleration. Steering ranges from-1 to 1, with steering >0 indicating a right turn, steering <0 indicating a left turn, and steering=0 indicating straight driving. In addition to the image, ego's current speed is {:.2f}m/s, yaw is {:.2f}. If there is no visible pedestrian in the image or the distance of the pedestrian is relatively far, ego should set the throttle =1.0 to accelerate. If ego's direction deviates from the current lane, steering adjustment is required. Finally, design reasonable throttle and steering output values for the ego based on the description of the image, the current speed and yaw. Please follow the following example to output content:\nDescribe:xxx \nthrottle:xxx \nsteer:xxx

Figure 13: The prompt of VLMs in the #JW scenario.

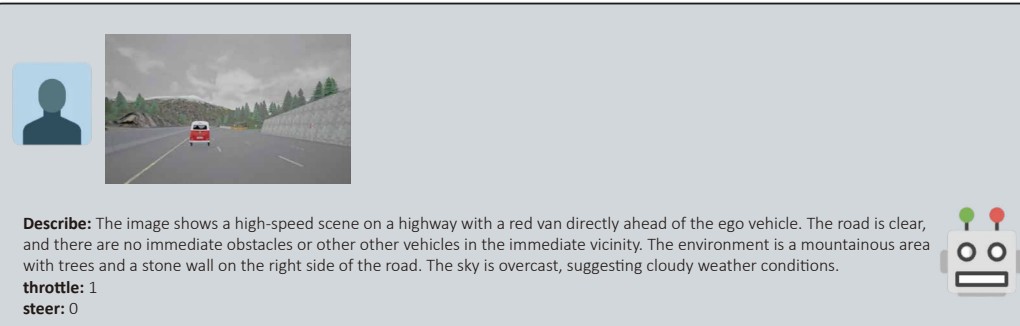

**Describe:** The image shows a high-speed scene on a highway with a red van directly ahead of the ego vehicle. The road is clear, and there are no immediate obstacles or other other vehicles in the immediate vicinity. The environment is a mountainous area with trees and a stone wall on the right side of the road. The sky is overcast, suggesting cloudy weather conditions.
**throttle:** 1
**steer:** 0

Figure 14: Example of VLMs inference #HW scenario.

**Adaptive action switching**: Each episode randomly selects a phase duration $T_{switch} \sim U[50, 400]$, alternating between visual RL exploration (high-entropy actions for broad state coverage) and VLMs execution (task-specific actions). This ensures asymptotically complete state-space visitation while preserving task coherence.

**Complementarity of policy distribution**: The off-policy feature of visual RL (e.g., DeepMDP) enables it to learn from historical data, while VLMs offer structured prior knowledge through language-visual alignment. The action distribution of the two satisfies the equation:

$$\mathcal{A}_{\text{optimal}} \subseteq \mathcal{S}(\pi_{\text{RL}}) \cup \mathcal{S}(\pi_{\text{VLM}}), \tag{18}$$

where, $\mathcal{S}(\cdot)$ is the set of actions included in the policy. The action support set of the collaborative policy encompasses the optimal action space, ensuring that $\mathcal{C}_V(\cdot)$ could learn more complete action-value function.

Table 7: Comprehensive experiments conducted in the #HW scenario validate the effectiveness of RAE. The best results for each metric are bolded.

| Idx. | ER ↑ | DD ↑ | Description |
|------|------|------|-------------|
| **M1** | 205 ± 141 | 215 ± 145 | $\mathcal{C}_V(\cdot)$ is generated by a pure VLMs policy without RL exploration |
| **M2** | 247 ± 135 | 263 ± 139 | $\mathcal{C}_V(\cdot)$ is generated by a pure RL policy without VLMs involved |
| **M3** | 174 ± 63 | 188 ± 67 | $\mathcal{C}_V(\cdot)$ is replaced by $\mathcal{C}_R(\cdot)$ |
| ERPV (Ours) | **299 ± 106** | **311 ± 109** | $\mathcal{C}_V(\cdot)$ is generated by RAE |

D.2 THE EXTRA EXPERIMENTS ON $\mathcal{C}_V(\cdot)$

To assess the effectiveness of the RAE method, we conducted a comparative experiment involving various pre-training methods for $\mathcal{C}_V(\cdot)$. The results are shown in Table 7 and can be analyzed in

several aspects: (1) **M1** demonstrates that training $\mathcal{C}_V(\cdot)$ solely with VLMs would lead the model to become trapped in a local optimum, resulting in an inability to evaluate actions with greater accuracy and comprehensiveness. (2) **M2** highlights the significance of a collaborative policy. When specific knowledge of VLMs is insufficient, the trained $\mathcal{C}_V(\cdot)$ remains suboptimal; consequently, its evaluation capability fails to encompass a broader action space. (3) **M3** illustrates the importance of the prior knowledge from the pre-trained $\mathcal{C}_V(\cdot)$. (4) Obviously, the RAE method for $\mathcal{C}_V(\cdot)$ can evaluate the action value of VLMs more accurately and comprehensively. Consequently, the performance of the learned policy is also optimal.

# E    ADDITIONAL EXPERIMENTAL RESULTS

## E.1    EXPERIMENTS ON CARRACING

CarRacing, a third-person perspective autonomous driving task, is well-suited for evaluating VLMs' reasoning capabilities in visually dense environments. We trained each benchmark in 100K steps and reported the metrics over 20 episodes with three different seeds. Table 8 presents the performance evaluation results on the CarRacing benchmark. The evaluation metrics is ER. The experimental results demonstrate that ERPV achieves the best performance compared to the previous methods, highlighting its broad applicability and potential. The VBE results show that the VLM has limited overall performance, with its suggested actions being reliable only in certain states. ERPV effectively identifies and leverages these locally reliable priors, thereby improving the sample efficiency of visual reinforcement learning.

Table 8: Testing performance comparison with SOTA methods on the CarRacing benchmark. The best results for each metric are bolded.

| Type | Methods | ER $\uparrow$ |
|---|---|---|
| Vanilla RL | SAC (Haarnoja et al., 2018b) | 287 ± 201 |
| | DeepMDP (Gelada et al., 2019) | 356 ± 169 |
| | Curl (Laskin et al., 2020a) | 462 ± 201 |
| | Drq (Kostrikov et al., 2020) | 354 ± 213 |
| | SPR (Schwarzer et al., 2020) | 380 ± 229 |
| | MLR (Yu et al., 2022) | 350 ± 186 |
| | PlayVirtual (Yu et al., 2021) | 83 ± 79 |
| | ResAct (Liu et al., 2025) | 503 ± 225 |
| Only VLMs | VBE (Mei et al., 2024) | -48 ± 28 |
| VLM-assisted RL | DSF (Xu et al., 2024b) | 497 ± 253 |
| | ASF (Zhou et al., 2024) | 453 ± 181 |
| | RSF | 338 ± 215 |
| | ERPV (Ours) | **576 ± 264** |

## E.2    EXPERIMENTS ON $\lambda_1$ AND $\lambda_2$

Table 9 and Table 10 present the results for parameters $\lambda_1$ and $\lambda_2$ across various values, respectively. The default settings are $\lambda_1=10$ and $\lambda_2=10$. The experiment was conducted using the control variable method.

In fact, $\lambda_1$ and $\lambda_2$ could be analyzed from the perspective of RL methodology. As shown in the Table 9 and Table 10, both should be set within a moderate range—either too large or too small values lead to degraded performance:

• $\lambda_1$ operates within the VPG module (corresponding to Equation (4)), controlling the strength of regularization that encourages consistency between the visual RL policy and the VLM's prior actions. The VLM provides helpful but suboptimal action priors. If $\lambda_1$ is too small, the guidance from the VLM is underutilized, limiting knowledge transfer. If $\lambda_1$ is too large, the policy may over-rely on the VLM's outputs, constraining the RL agent's ability to discover superior policies through exploration.

- $\lambda_2$ acts in the VER module (corresponding to Equation (7)), influencing the adaptation of the temperature coefficient in SAC, thereby indirectly regulating policy entropy and exploration. If $\lambda_2$ is too small, the temperature coefficient decays slowly, maintaining high entropy and excessive exploration in later stages, which harms stability. Conversely, if $\lambda_2$ is too large, the temperature coefficient drops rapidly, prematurely suppressing exploration and leading to insufficient data diversity, which undermines learning effectiveness.

Table 9: Testing performance of different $\lambda_1$ in the #HW scenario, where $\lambda_2$=10.0. The best result of each metric is bolded.

| Metric/Parameters | $\lambda_1$ | | |
|---|---|---|---|
| | 5.0 | 10.0 | 15.0 |
| ER↑ | $217 \pm 90$ | $\mathbf{299 \pm 106}$ | $246 \pm 104$ |
| DD↑ | $230 \pm 94$ | $\mathbf{311 \pm 109}$ | $235 \pm 101$ |

Table 10: Testing performance of different $\lambda_2$ in the #HW scenario, where $\lambda_1$=10.0. The best results of each metric are bolded.

| Metric/Parameters | $\lambda_2$ | | | |
|---|---|---|---|---|
| | 5.0 | 10.0 | 15.0 | 20.0 |
| ER↑ | $251 \pm 101$ | $\mathbf{299 \pm 106}$ | $297 \pm 129$ | $288 \pm 95$ |
| DD↑ | $264 \pm 104$ | $\mathbf{311 \pm 109}$ | $307 \pm 132$ | $300 \pm 99$ |

We also tested hyperparameters across different tasks, evaluating the performance of $\lambda_1$ and $\lambda_2$ at various values in the Cartpole, Swingup task. The results are presented in Table 11.

Table 11: Testing performance of different $\lambda_1$ and $\lambda_2$ in the Cartpole, Swingup task. The best results of each metric are bolded.

| Metric/Parameters | $\lambda_1=\lambda_2$ | | | |
|---|---|---|---|---|
| | 5.0 | 10.0 | 15.0 | 20.0 |
| ER↑ | $833\pm 3$ | $\mathbf{855 \pm 2}$ | $844 \pm 4$ | $812 \pm 3$ |

These experimental results demonstrate that the hyperparameters of ERPV, where $\lambda_1=\lambda_2$=10, also remain relatively robust across different tasks. It suggests that our method is not sensitive to fine-tuning, which facilitates deployment in new environments. Therefore, other researchers aiming to apply this method in other environments can initially use the hyperparameter settings presented in this paper.

### E.3 HYBRID VLM-RL TRAJECTORY

We implemented DeepMDP (Gelada et al., 2019) with hybrid VLM-RL trajectory interactions, using two modes: (1) The VLM interaction frequency tied to training steps, and (2) full-episode VLM interaction with adjustable trajectory proportion. We conducted thorough experiments in the #HW scenario in Carla, and the results are shown in Table 12 and Table 13. Given the imperfect VLM, hybrid interaction may be suboptimal, leading to suboptimal RL. In contrast, ERPV leverages reliable VLM actions to directly guide RL policy learning; this prevents unreliable knowledge propagation.

### E.4 COMPUTE RESOURCES

The graphics card used in this paper is one 80GB PCIe of NVIDIA A100. The training cost of different models is shown in Table 14. It includes the graphics card memory (GCM) occupied during training phase, the average training time (Avg. TT) and the average reasoning time (Avg. RT) for testing 10 inferences. RL combined with VLM will definitely slow down, but ERPV performs better than other methods.

Table 12: Testing performance of the VLM interaction frequency tied to training steps. The best results of each metric are bolded.

| Metric/Training steps | 0(DeepMDP) | 2.0 | 5.0 | 10.0 | 15.0 | 20.0 | ERPV |
|---|---|---|---|---|---|---|---|
| ER↑ | 100 ± 68 | 145 ± 94 | 171 ± 77 | 173 ± 67 | 141 ± 97 | 167 ± 89 | **299 ± 106** |
| DD↑ | 126 ± 77 | 158 ± 97 | 188 ± 79 | 189 ± 69 | 157 ± 102 | 181 ± 93 | **311 ± 109** |

Table 13: Testing performance of the full-episode VLM interaction with adjustable trajectory proportion. The best results of each metric are bolded.

| Metric/Trajectory ratio (%) | 0(DeepMDP) | 10.0 | 20.0 | 50.0 | 80 | ERPV |
|---|---|---|---|---|---|---|
| ER↑ | 167 ± 89 | 154 ± 68 | 143 ± 80 | 141 ± 81 | 136 ± 50 | **299 ± 106** |
| DD↑ | 181 ± 93 | 168 ± 70 | 154 ± 82 | 150 ± 82 | 147 ± 53 | **311 ± 109** |

Table 14: The resource consumption of different model sizes

| Model | GCM (M) | Avg. TT (hours) | Avg. RT (second) |
|---|---|---|---|
| DeepMDP (Gelada et al., 2019) | ≈ 3366 | 8.5 | 0.0012 ± 0.0003 |
| SAC (Haarnoja et al., 2018b) | ≈ 3366 | 8.3 | 0.0012 ± 0.0003 |
| DSF (Lee et al., 2025) with Qwen2-VL-7B | ≈ 20971 | 34.3 | 6.95 ± 0.76 |
| ASF (Zhou et al., 2024) with Qwen2-VL-7B | ≈ 20971 | 25.9 | 6.95 ± 0.76 |
| ERPVwith Qwen2-VL-7B | ≈ 20971 | 26.6 | 6.95 ± 0.76 |
| ERPVwith Qwen2-VL-2B | ≈ 7060 | 25.4 | 4.94 ± 0.61 |
| ERPVwith LLava-1.5-7B | ≈ 14722 | 23.5 | 3.66 ± 1.03 |

## E.5 EXPERIMENTS ON HARD TASKS

We have systematically evaluated DMControl tasks in Table 2, covering high-dimensional action spaces and complex dynamics (e.g., Walker and Cheetah). As a standard benchmark for simulating real-world robotic control, DMControl poses significantly higher difficulty than conventional RL environments.

Additionally, we further selected the hard task in DMControl, called Quadruped, which features extremely high action dimensionality (12 Dimension) and more complex dynamics. The experiment followed the RAD (Laskin et al., 2020b) setup, with 500K training steps and 10 episodes of evaluation. The results are shown in Table 15. ERPV improves the learning efficiency of RL by using VLM in some states to provide higher-value prior action guidance, and its performance can surpass that of base RL. The results further demonstrates its effectiveness in more high-dimensional and complex dynamic tasks.

| Models | Quadruped, Walk | Quadruped, Run |
|---|---|---|
| RAD (Laskin et al., 2020b) | 246 ± 108 | 199 ± 177 |
| TACO (Zheng et al., 2023) | 345 ± 89 | 366 ± 40 |
| MaDI (Grooten et al., 2024) | 277 ± 92 | 265 ± 36 |
| ResAct (Liu et al., 2025) | 385 ± 81 | 377 ± 76 |
| Dreamer V3 (Hafner et al., 2025) | 353 ± 27 | 331 ± 42 |
| VBE | 104 ± 199 | 103 ± 199 |
| ERPV (Ours) | **400 ± 93** | **380 ± 95** |

Table 15: Testing comparison with SOTA methods of hard tasks in DMControl. The best results of each metric are bolded.

### E.6 EXPERIMENTS ON OVERFITTING

Our method focuses on real-time control, so VLM is not available in the test. To evaluate overfitting, we tested the model weights trained in the HW scenario in the following two situations:

(1) For different weather tests, we changed the weather in the HW scene to rain to modify the visual input. The results are shown in Table 16.

| Model \Scenario (Metrics) | HW (ER↑, DD↑) | HW with rain (ER↑, DD↑) |
|---|---|---|
| DeepMDP (Gelada et al., 2019) | 86 ± 43, 105 ± 50 | 2 ± 3, 14 ± 5 |
| ERPV (Ours) | **299 ± 106, 311 ± 109** | **83 ± 38, 129 ± 52** |

Table 16: Testing comparison of overfitting for different weather. The best results of each metric are bolded.

(2) Cross-scenario testing. The results are shown in Table 17.

| Model \Scenario (Metrics) | HW (ER↑, DD↑) | JW (ER↑, DD↑) |
|---|---|---|
| DeepMDP (Gelada et al., 2019) | 86 ± 43, 105 ± 50 | 9 ± 2, 10 ± 2 |
| ERPV (Ours) | **299 ± 106, 311 ± 109** | **37 ± 12 38 ± 12** |

Table 17: Testing comparison of overfitting for cross-scenario. The best results of each metric are bolded.

The results show that it still outperforms the basic RL, confirming that there is no excessive overfitting.

## F FURTHER DISCUSSION ABOUT VPG AND VER MODULES

### F.1 DIFFERENT ROLES

This paper primarily introduces two innovative components: the VPG and VER modules. VPG and VER plays different roles and they are not conflict:

• VPG chooses the reliable VLM actions and then learns the policy network from these filtered VLM actions. Hence, it only works in partial or specific states.

• VER evaluates the reliability of the current policy by comparing it with VLM actions and provides global entropy control of all states. Then, the global entropy is used to decide between exploitation and exploration during learning.

• If VLM actions at some states are judged reliable by VPG and our policy network has converged to the VLM actions, RL will prefer to exploit under the influence of VER. Otherwise, VER encourages exploration to attempt diverse actions.

During the learning process, SAC balances exploration and exploitation through global entropy, while VER can avoid situations where reliable actions filtered out by VPG still have excessive entropy. Therefore, these two modules are not contradictory, and they can better balance exploration and exploitation than vanilla RL.

### F.2 DISCUSSION ON FAILURE MODES AND ROBUSTNESS

A critical consideration for any method leveraging partially reliable external knowledge is its behavior when that knowledge is systematically erroneous. We now discuss the failure modes of ERPV and its inherent recovery mechanisms.

(1) The weak inference mode of VLM. In a worst-case scenario, the VLM might provide consistently suboptimal or harmful actions for a particular state or set of states (e.g., always suggesting a sharp

turn in a straight lane due to a training bias). In the experiment shown in Table 4, the performance of Qwen2-VL-2B is even worse than that of the random strategy, but the performance of ERPV can be maintained at a level comparable to that of base RL approach.

(2) The catastrophic inference mode of VLM (completely ineffective). We introduce the Worst Model on Carla, which provides unreliable guidance in all states. As shown in Table 4, ERPV maintains performance on par with the base RL approach.

The robustness of ERPV stems from the adaptive gating mechanism in the VPG module. For each state, VPG computes the advantage of the VLM-suggested action over the current policy: $\mathcal{A}_v = Q_v - Q_r$. When the VLM consistently performs worse than the current policy, $\mathcal{A}_v$ becomes non-positive: $Q_f = \mathbf{Max}(0, \mathcal{A}_v)$. It automatically suppresses the supervisory signal from the VLM. In such states, the RL agent disregards the VLM's guidance entirely and reverts to standard SAC updates, learning solely from environmental rewards.

It is worth noting that in extreme cases where the VLM prior is entirely ineffective, while the VPG module could automatically suppress harmful guidance via its mechanism, the VER module may still compute a large state transition discrepancy $\text{Dist}[\mathcal{P}(s_{t+1,i}|s_{t,i}, a_{r,i}), \mathcal{P}(\tilde{s}_{t+1,i}|s_{t,i}, a_{v,i})]$ due to the VLM's consistently poor actions. This results in persistently low confidence $\mathcal{G}$, potentially trapping the policy in a high-entropy exploration regime. Since the VLM-induced state predictions diverge significantly from actual transitions, using it as a signal for entropy regulation becomes counterproductive. In such scenarios, it is advisable to reduce $\lambda_2$ or even deactivate the VLM's influence in the VER module.

## F.3 QUALITATIVE ANALYSIS

This method is based on the heuristic design of VLM and in fact cannot be strictly proved through theoretical analysis. However, qualitative analysis can be conducted through the policy improvement of RL:

Let $Q_{r,k}$ denote the $Q$-value learned by RL at the $k$-th iteration, and $Q_v$ represent the $Q$-value computed by the VLM. We analyze the following typical learning situations:

(1) When $Q_{r,k+1} \geq Q_v$ and $Q_{r,k} \geq Q_v$, the system operates in conventional RL learning mode since the $\mathcal{L}\_\mathcal{V}$ in Eq. 4 is not constrained. The policy improvement guarantees the relationship $Q_{r,k+1} \geq Q_{r,k}$.

(2) When both $Q_{r,k+1} \geq Q_v$ and $Q_{r,k} < Q_v$ are satisfied, the RL policy's effectiveness can still be validated through the inequality $Q_{r,k+1} \geq Q_{r,k}$.

(3) The situation that $Q_{r,k+1} < Q_v$ and $Q_{r,k} > Q_v$ doesn't exist. This is because the learning of RL has ensured that $Q_{r,k+1} \geq Q_{r,k}$.

(4) As long as cases (1) nor (2) exist at some $k$, the learning of RL can break through $Q_v$ and become pure RL optimality learning. If all satisfy $Q_{r,k+1} < Q_v$ and $Q_{r,k} < Q_v$ for any $k$, this indicates that the $Q$-value of RL consistently underperforms compared to the $Q$-value of VLM, reflecting that $\mathcal{L}\_\mathcal{V}$ in Eq. 4 plays a leading role. However, due to bounded weight constraints, actions, and Q-value normalization operations, this situation generally does not occur in practice. We systematically analyzed the gradient magnitudes of $\mathcal{L}\_\pi$ and $\mathcal{L}\_\mathcal{V}$ in Eq. 4. The gradient magnitude of $\mathcal{L}\_\pi$ ranges from 0.5 to 2.5, while that of $\mathcal{L}\_\mathcal{V}$ is confined to [0.02, 0.05]. Given that $\mathcal{L}\_\mathcal{V}$'s gradient magnitude is significantly lower than $\mathcal{L}\_\pi$, $\mathcal{L}\_\mathcal{V}$ does not dominate the gradient updates during training.

## F.4 DISCUSSION ON THE EXPLORATION OF ERPV

ERPV does not explore indefinitely for the following reasons:

1.The entropy coefficient $\alpha$ is constrained by an auxiliary regularization term (Eq. 7). We do not enforce excessive exploration and preserve the baseline exploration level of vanilla SAC.

2.The target $\bar{\alpha}$ introduced in Eq. 7 is bounded, inherently preventing unbounded exploration. Fig. 5(b) shows that $\alpha$ remains stable throughout training.

3.To further demonstrate robustness under highly unreliable VLM guidance, we report the change of $\alpha$ on the Cheetah-Run task in Table 18, where the VLM performs poorly (reward: $5 \pm 3$). Even in this case, $\alpha$ fluctuates within a reasonable range without diverging.

| Train Step \Methods | SAC (Haarnoja et al., 2018b) | DeepMDP (Gelada et al., 2019) | ERPV |
|---|---|---|---|
| 0 | 0.1 | 0.1 | 0.1 |
| 10,000 | 0.042 | 0.062 | 0.066 |
| 30,000 | 0.020 | 0.052 | 0.073 |
| 60,000 | 0.036 | 0.056 | 0.069 |
| 80,000 | 0.039 | 0.055 | 0.064 |
| 100,000 | 0.038 | 0.052 | 0.060 |

Table 18: The change of $\alpha$ on the Cheetah-Run task.

In addition, although ERPV does encourage its utilization when multiple operations are reliable, it is rare in practice for two actions to be reliable but diverse simultaneously. The reason is that, under the influence of the VPG module, the RL strategy is guided toward the reliable actions provided by the VLM, causing their values to align in certain states and generate higher cumulative rewards rather than maintaining high action diversity.

### F.5 VISUAL ANALYSIS OF VPG

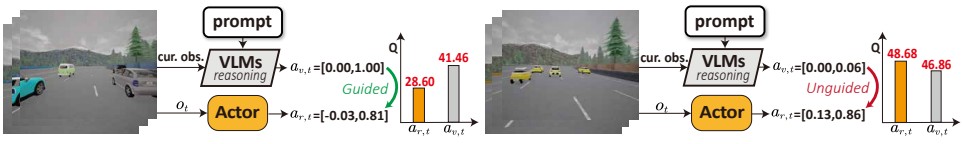

(a) The superior $Q$-value of $a_{v,t}$ over $a_{r,t}$.  (b) The superior $Q$-value of $a_{r,t}$ over $a_{v,t}$.

Figure 15: Visual Analysis of VPG in Carla.

Fig. 15 (a) and (b) analyzes $Q$-value dynamics between VLMs and visual RL under different complex scenarios during the training process. $\mathbf{a}_t = [steer, throttle] \in [-1,1]$ with $steer$ controls turning direction ($< 0$:left, 0:straight, $> 0$:right) and $throttle$ regulates speed ($\leq 0$:brake/decel, $> 0$:accel). Note that the action values produced by VLMs are not discrete, such as acceleration or deceleration. Instead, VLMs generate continuous action value reasoning through the design prompt. Analysis can be obtained: (a) Straight-line Scenario: Since traveling in a straight line is risk-free and yields higher returns, even in the presence of surrounding vehicles, $\mathcal{C}_V(o_t, \mathbf{a}_{v,t} = [0.00, 1.00]) = 41.46$ outperforms $\mathcal{C}_R(o_t, \mathbf{a}_{r,t} = [-0.03, 0.81]) = 28.60$. This result activates the constraint in Equation (3) ($Q_f > 0$). (b) Obstacles-Ahead Scenario: This scenario illustrates the limitations of predictions for VLMs, where speeding straight ahead increases the likelihood of a collision, in contrast to RL's converged policy ($Q_v < Q_r$), which results in $Q_f = 0$ without the need for constraints. The analysis of these scenarios visually demonstrates the effectiveness of the VPG evaluation module, which can enhance the policy performance of the visual RL method.

## G LIMITATIONS AND FUTURE WORKS

Although our method is based on VLMs heuristic reinforcement learning, it is difficult to carry out strict theoretical proof. But several directions remain open. First, the performance of current VLMs degrades in novel, complex perceptual scenarios, highlighting the need for fine-tuned, task-specialized small VLMs to improve generalization. Second, the fixed-frequency VLM querying and inference quality can be enhanced via multi-frame prompting, model compression, or adaptive invocation policies learned by RL. Finally, while our modules use heuristic regularization (e.g., VER weight), future work could explore self-adjusting mechanisms that respond to runtime signals like consistency drops.

