# OpenReview forum: "ERPV: Enhancing Visual Reinforcement Learning with Partially Reliable Knowledge from VLMs"
_ICLR.cc/2026/Conference — Submitted to ICLR 2026_

### Official Review · Reviewer_ZmUU · 2025-10-27

**Soundness:** 3
**Presentation:** 3
**Contribution:** 3
**Rating:** 6
**Confidence:** 4

**Summary:**

This paper introduces ERPV, a novel visual reinforcement learning method.
Specifically, ERPV leverages the vision language model to (1) select actions to guide policy learning and (2) provide reference actions for better exploration.
Experiment results on various benchmarks show that ERPV achieve superior performance and learning efficiency.

**Strengths:**

1. The paper writing and structure are clear.
2. Good motivation and novelty regarding VLM as the prior knowledge for policy learning and exploration (VPG and VER components).
3. Empirical results demonstrate the superior improvements on various benchmarks.

**Weaknesses:**

1. As mentioned in the limitation, it would be great to analyse the theoretical properties of ERPV, including convergence and optimality. It feels like be easy to extend from SAC.
2. The training speed deeply depends on the inference speed of VLM. It would be better to have the statistics of the training speed comparison in section 4.4.

**Questions:**

See weakness.

---

> ### Author Response · Authors · 2025-11-26
>
> Thank you very much for the time and effort you have devoted to reviewing our work. We are grateful for your positive comments and encouraged by your feedback. We address the points raised in the comments as follows:
>
> **W1**
>
> This method is based on the heuristic design of VLM and in fact cannot be strictly proved through theoretical analysis. However, qualitative analysis can be conducted through the policy improvement of RL [1]:
>
> Let $Q_{r, k}$ denote the $Q$-value learned by RL at the $k$-th iteration, and $Q_{v}$ represent the $Q$-value computed by the VLM. We analyze the following typical learning situations:
>
> (1) When $Q_{r, k+1} \ge Q_{v}$ and $Q_{r, k} \ge Q_{v}$, the system operates in conventional RL learning mode since the $\mathcal{L}\_{\mathcal{V}}$ in Eq. 4 is not constrained. The policy improvement guarantees the relationship $Q_{r, k+1} \ge Q_{r, k}$.
>
> (2) When both $Q_{r, k+1} \ge Q_{v}$ and $Q_{r, k} < Q_{v}$ are satisfied, the RL policy's effectiveness can still be validated through the inequality $Q_{r, k+1} \ge Q_{r, k}$.
>
> (3) The situation that $Q_{r, k+1} < Q_{v}$ and $Q_{r, k} > Q_{v}$ doesn't exist. This is because the learning of RL has ensured that $Q_{r, k+1} \ge Q_{r, k}$.
>
> (4) As long as cases (1) nor (2) exist at some $k$, the learning of RL can break through $Q_{v}$ and become pure RL optimality learning. If all satisfy $Q_{r, k+1} < Q_{v}$ and $Q_{r, k} < Q_{v}$ for any $k$, this indicates that the $Q$-value of RL consistently underperforms compared to the $Q$-value of VLM, reflecting that $\mathcal{L}\_{\mathcal{V}}$ in Eq. 4 plays a leading role. However, due to bounded weight constraints, actions, and Q-value normalization operations, this situation generally does not occur in practice. We systematically analyzed the gradient magnitudes of $\mathcal{L}\_{\mathcal{\pi}}$ and $\mathcal{L}\_{\mathcal{V}}$ in Eq. 4. The gradient magnitude of $\mathcal{L}\_{\mathcal{\pi}}$ ranges from 0.5 to 2.5, while that of $\mathcal{L}\_{\mathcal{V}}$ is confined to [0.02, 0.05]. Given that $\mathcal{L}\_{\mathcal{V}}$'s gradient magnitude is significantly lower than $\mathcal{L}\_{\mathcal{\pi}}$, $\mathcal{L}\_{\mathcal{V}}$ does not dominate the gradient updates during training.
>
> Relevant theoretical analysis will be added in Section F of Appendix. Thank you!
>
>
> **W2**
>
> The graphics card used in this paper is an 80GB PCIe NVIDIA A100. We have reported the inference speed and training memory usage of different VLMs in the Section E.4 of Appendix. We report the average training cost for different models in Table 4:
>
> | Models                 | Training speed (hours) | Memory (M)      |
> | ---------------------- | ---------------------- | --------------- |
> | DeepMDP [2]                | $\approx$ 8.5          | $\approx$ 3366  |
> | ERPV with Qwen2-VL-7B  | $\approx$ 26.6         | $\approx$ 20971 |
> | ERPV with Qwen2-VL-2B  | $\approx$ 25.4         | $\approx$ 7060  |
> | ERPV with LLava-1.5-7B | $\approx$ 23.5         | $\approx$ 14722 |
>
> RL combined with VLM will definitely slow down, but ERPV performs better than other methods. We would revise the content of Section E.4 of Appendix to more comprehensively compare the training cost of the training process. Thank you!
>
> **References**
>
> [1] Tuomas Haarnoja, Aurick Zhou, Kristian Hartikainen, George Tucker, Sehoon Ha, Jie Tan, Vikash Kumar, Henry Zhu, Abhishek Gupta, Pieter Abbeel, et al. Soft actor-critic algorithms and applications. *arXiv preprint arXiv:1812.05905*, 2018b.
>
> [2] Carles Gelada, Saurabh Kumar, Jacob Buckman, Ofir Nachum, and Marc G Bellemare. Deepmdp: Learning continuous latent space models for representation learning. In International conference on machine learning*, pp. 2170–2179. PMLR, 2019.

---

> > ### Comment · Reviewer_ZmUU · 2025-11-27
> >
> > Thank you for your response. I maintain a positive view of the paper.

---

### Official Review · Reviewer_cNdj · 2025-10-29

**Soundness:** 3
**Presentation:** 3
**Contribution:** 2
**Rating:** 6
**Confidence:** 3

**Summary:**

The paper proposes ERPV as an approach to benefit from common sense in VLM in visual reinforcement learning (RL). ERPV leverages predictions frmo a VLM  in a soft-actor-critic RL framework  by regerresing the VLM action. The regression loss is weighted by a coefficient representing the advantage of using VLM-based actions over using action from the RL policy. The authors also propose using the prediction error of a dynamics model conditioned on the VLM action as an exploration signal.

**Strengths:**

- The paper is well written
- Leveraging VLM common sense to guide policy search is a novel and interesting idea
- The results are strong and quite promising
- Experiments include ablations of various design choices and clearly disentangle the role of different components

**Weaknesses:**

- The paper lacks an extensive discussion of the main assumption made here: VLMs are trained with action-free data, the best they could actually do in action selection is either some sort of nearest neighbor if the domain data was seen during VLM training, or provide actions that are at a semantic level at best. Most action representations in complex systems (e.g. robotic manipulation) are far more challenging on the lower (non-semantic) level. In such complex environments, it is unclear how such a supervision could even help beyond just simple early-state exploration ( a problem that is potentially non-existent if expert data is available, which is becoming the case)
- Most environments used in the experiments are low-dimensional, and involve simple dynamics, it would be interesting to demonstrate the applicability of the method to more complex domains, or at least understand its limits in such domain

**Questions:**

Can you provide a more extensive discussion on how you expect your assumption of "VLM actions being a good source of supervision" to scale or fail to scale to high-dimensional complex tasks that involve low-level action?

---

> ### Author Response · Authors · 2025-11-26
>
> We are truly grateful for your thoughtful remarks and experimental suggestions. These remarks shed light on what we can improve and are crucial for refining our work. We address your main concerns as follows:
>
> **W1 & Q**
>
> 1.As you said, VLMs are not completely reliable when outputting specific actions, and this is also the motivation of our method. In our work, VLMs may not be a good source of supervision. That is to say, it could be partially reliable or even very unreliable (The reliability experiments of VLM to different degrees are shown in Table 4).
>
> 2.Although VLMs are not entirely reliable, conventional RL start with completely random strategies, so it is helpful to draw inspiration from VLMs.  Although VLMs may be not trained with a large number of image-action pairs related to Carla, due to the powerful generalization reasoning ability of VLMs, it can also infer better prior actions than random actions in certain states. For instance, in autonomous driving tasks, when given a simple image of a straight road, VLMs with appropriate prompts can reliably output "accelerated straight line". Our approach combines the powerful generalization reasoning ability of VLMs with the later self-exploration and development of RL. When VLM infers better actions than the exploration actions of RL in certain states, the learning efficiency of RL policies can be accelerated through guidance.
>
> 3.When there is expert data, VLM is indeed not necessary. However, for some tasks, collecting expert data (such as operating some uncommon machines) is very expensive. VLM provides solutions to handle such situations. In Figure 6, Figure 14 and Figure 15, we also provide visual examples, which can help readers intuitively understand the role that the reasoning actions of VLMs play in certain states.
>
>
>
> **W2**
>
> 1.We have systematically evaluated DMControl tasks in Table 2, covering high-dimensional action spaces and complex dynamics (e.g., Walker and Cheetah). As a standard benchmark for simulating real-world robotic control, DMControl poses significantly higher difficulty than conventional RL environments.
>
> 2.Additionally, we further selected the hard task in DMControl, called **Quadruped**, which features extremely high action dimensionality (12 Dimension) and more complex dynamics. The experiment followed the RAD [1] setup, with 500K training steps and 10 episodes of evaluation. The results shows in the following Table. ERPV improves the learning efficiency of RL by using VLM in some states to provide higher-value prior action guidance, and its performance could surpass that of base RL methods. The results further demonstrates its effectiveness in more high-dimensional and complex dynamic tasks.
>
> |                     | Quadruped, Walk | Quadruped, Run |
> | ------------------- | --------------- | -------------- |
> | **Method**          | **ER**          | **ER**         |
> | RAD [1]             | 246 ± 108       | 199 ± 177      |
> | TACO [2]            | 345 *±* 89      | 336 ± 40       |
> | MaDi [3]            | 277 *±* 92      | 265 ± 36       |
> | ResAct [4]          | 385 ± 81        | 377 ± 76       |
> | Dreamer V3 [5]      | 353 ± 27        | 331 ± 42       |
> | VLMs-based Executor | 104 ± 199       | 103 ± 199      |
> | ERPV                | **400 ± 93**    | **380 ± 95**   |
>
> The relevant experiments will be added in Section E.5 of the Appendix. Thank you!
>
>
>
> **References**
>
> [1] Laskin, Misha, et al. "Reinforcement learning with augmented data." Advances in neural information processing systems 33 (2020): 19884-19895.
>
> [2] Zheng, Ruijie, et al. "TACO: Temporal Latent Action-Driven Contrastive Loss for Visual Reinforcement Learning." Advances in Neural Information Processing Systems 36 (2023): 48203-48225.
>
> [3] Grooten, Bram, et al. "Madi: Learning to mask distractions for generalization in visual deep reinforcement learning." arXiv preprint arXiv:2312.15339 (2023).
>
> [4] Liu, Zhenxian, Peixi Peng, and Yonghong Tian. "Visual Reinforcement Learning with Residual Action." Proceedings of the AAAI Conference on Artificial Intelligence. Vol. 39. No. 18. 2025.
>
> [5] Hafner, D., Pasukonis, J., Ba, J. et al. Mastering diverse control tasks through world models. Nature 640, 647–653 (2025).

---

### Official Review · Reviewer_K2Jp · 2025-10-31

**Soundness:** 2
**Presentation:** 2
**Contribution:** 2
**Rating:** 4
**Confidence:** 2

**Summary:**

This paper presents ERPV, a method that integrates partially reliable knowledge from Vision-Language Models (VLMs) into visual reinforcement learning (VRL). The key motivation is that while VLMs contain useful commonsense priors, their action reasoning is often unreliable or inconsistent across states. To address this, the authors propose two mechanisms: Value-aware Policy Guidance (VPG): which dynamically estimates the reliability of VLM-inferred actions by comparing their Q-values with those of the RL policy, selectively applying guidance when the VLM’s suggestion appears better. VLM-guided Entropy Regularization (VER): adjusts exploration via an entropy coefficient that depends on how consistent the RL policy’s actions are with the VLM’s inferred actions, encouraging exploration when they diverge and exploitation when they align. Experiments across Carla, DMControl, and CarRacing benchmarks show that ERPV improves both sample efficiency and final performance, even when the VLM guidance is noisy or unreliable.

**Strengths:**

- Timely and relevant problem: The paper addresses an emerging and underexplored question — how to integrate large pretrained vision-language models into reinforcement learning while handling their imperfect reasoning.

- Thoughtful formulation: The introduction of reliability estimation (VPG) and entropy adjustment (VER) feels intuitive and conceptually clean, combining the strengths of teacher–student distillation and adaptive exploration control.

- Robust empirical results: Across several benchmarks, ERPV consistently outperforms prior VLM-assisted RL baselines (e.g., DSF, ASF, DGC), and even performs comparably to base RL when VLMs are unreliable.

- Comprehensive experiments: The inclusion of diverse settings (e.g., CARLA, DMControl), multiple VLM backbones (Qwen2-VL, LLava), and ablations on both modules (VPG/VER) provides solid empirical evidence.

- Clear presentation: Figures and tables (especially Fig. 5 showing dynamics and entropy coefficient) make it easy to follow how the method behaves during training.

**Weaknesses:**

- Incremental conceptual novelty: The main novelty lies in combining selective guidance and entropy modulation, but both components resemble ideas from adaptive distillation and uncertainty-aware exploration. The conceptual leap is moderate.

- Lack of theoretical insight: The paper would benefit from some formal justification (e.g., why the proposed difference or transition-based confidence metric leads to stable convergence).

- Dependence on pretrained critic: Since ​the critic is trained using VLM actions, its generalization and possible bias are underexplored — what happens if the pretraining domain diverges from the RL environment?

- Limited real-world validation: All experiments are simulation-based. The authors’ claim of “real-time deployability” is interesting, but there’s no demonstration on an actual robotic platform.

- Scalability & compute details: The training cost of ERPV compared to vanilla SAC or other VLM-distilled methods isn’t discussed, making it hard to assess practicality.

**Questions:**

-How sensitive is ERPV to the hyperparameters lambda? Does over-weighting VPG risk reinforce VLM errors?

- Could the proposed Action Confidence Function (ACF) be replaced by simpler distance metrics (e.g., cosine similarity between action logits) without major loss?

- How much does the performance depend on the choice or pretraining quality of the VLM critic?

- Have you tested ERPV with textual prompts that are deliberately ambiguous or wrong to examine robustness?

- Could the approach be generalized to multi-modal feedback beyond actions, such as state representations or reward shaping?

---

> ### Author Response · Authors · 2025-11-23
>
> Thank you very much for your time and effort in reviewing our work. We truly appreciate your constructive and insightful feedback. Below, we provide detailed responses to the **Weaknesses** raised in your review:
>
> **W1**
>
> Our work focuses on integrating VLMs with RL, aiming to leverage partially reliable knowledge from VLMs to enhance RL policies. Our method is specifically designed by jointly exploiting the characteristics of both VLMs and RL, effectively addressing limitations in prior VLM-to-RL knowledge distillation methods (e.g., DSF and ASF, as evaluated in our experiments).
>
>
>
> **W2**
>
> From the perspective of learning mechanisms, transition samples inherently contain richer contextual information, since transitions encode both spatial features ([9,128,128]) and implicit dynamic interaction patterns through temporal correlations. In contrast, actions exist merely as low-dimensional numerical vectors, lacking semantic abstraction capabilities for complex scenarios. This viewpoint has already been mentioned in line 258 of the manuscript.
>
>
> **W3**
>
> 1.Due to the unknown pre-training domain of large models, we cannot assess the degree of overlap or potential relationships between the current RL task domain and the pre-training domain.
>
> 2.To validate the generalization capability of the pre-training methodology for the VLM-critic network, we conducted experiments under extreme conditions where the VLM inference results were completely unreliable (see Table 2 and Table 4). The experimental results show that even on the worst-performing pre-training model, the proposed method can still maintain the performance not lower than that of base RL. This result fully demonstrates the robustness and generalization of the proposed method.
>
>
>
> **W4**
>
> The "real-time deployability" stems from the lightweight RL policy design: inference latency per step is 0.0012 ± 0.0003 seconds. We follow the standard validation paradigm in most RL research (e.g., [4]) by conducting experiments in simulation environments. Future work will focus on real-world deployment. Thank you!
>
>
>
> **W5**
>
> The graphics card used in this paper is an 80GB PCIe NVIDIA A100. Section E.4 of Appendix reports only the GPU memory usage of ERPV with different large models during training. For Qwen2-VL-7B, we further clarify the comparison with VLM-distilled methods and Vanilla SAC:
>
> | Method          | train time (hours) | Memory (M) | ER, DD                    |
> | --------------- | ------------------ | ---------- | ------------------------- |
> | DSF [1]         | $\approx $ 34.3              | $\approx $20971     | 208 ± 82，220 ± 86        |
> | ASF [2]         | $\approx $ 25.9              | $\approx $20971     | 207 ± 86， 221 ± 90       |
> | Vanilla SAC [3] | $\approx $ 8.3               | $\approx $3366      | 86 ± 43, 105 ± 50         |
> | ERPV            | $\approx $ 26.6              | $\approx $ 20971     | **299 ± 106， 311 ± 109** |
>
> RL combined with VLM will definitely slow down, but ERPV performs better than other methods. We would revise the content of Section E.4 of Appendix to more comprehensively compare the training cost of the training process.
>
>
>
> **References**
>
> [1] Yi Xu, Yuxin Hu, Zaiwei Zhang, Gregory P Meyer, Siva Karthik Mustikovela, Siddhartha Srinivasa, Eric M Wolff, and Xin Huang. Vlm-ad: End-to-end autonomous driving through vision-language model supervision. *arXiv preprint arXiv:2412.14446*, 2024b.
>
> [2] Zihao Zhou, Bin Hu, Chenyang Zhao, Pu Zhang, and Bin Liu. Large language model as a policy teacher for training reinforcement learning agents. In *Proceedings of the Thirty-Third International Joint Conference on Artificial Intelligence*, pp. 5671–5679, 2024.
>
> [3] Tuomas Haarnoja, Aurick Zhou, Kristian Hartikainen, George Tucker, Sehoon Ha, Jie Tan, Vikash Kumar, Henry Zhu, Abhishek Gupta, Pieter Abbeel, et al. Soft actor-critic algorithms and applications. *arXiv preprint arXiv:1812.05905*, 2018b.
>
> [4] Danijar Hafner, Jurgis Pasukonis, Jimmy Ba, and Timothy Lillicrap. Mastering diverse control tasks through world models. *Nature*, pp. 1–7, 2025

---

> ### Author Response · Authors · 2025-11-23
>
> Thank you very much for your time and effort in reviewing our work. We truly appreciate your constructive and insightful feedback. Below, we provide detailed responses to the **Questions** raised in your review:
>
>
>
> **Q1**
>
> Extensive experiments on different $\lambda_1$ and $\lambda_2$ values are presented in Appendix E.2. Specifically, we conducted systematic hyperparameter studies on the Carla and DMControl tasks, including analysis of underfitting (too small) and overfitting (too large) regimes. Both tasks use the same set of hyperparameters to verify the universality and robustness of the method. Thank you!
>
>
>
> **Q2**
>
> In the ablation study (Table 3, M4 and M5), we evaluated alternative computation schemes for ACF, including the direct distance between action values. The performance comparison confirms the effectiveness of our transition-based ACF computation approach. Thank you!
>
>
>
> **Q3**
>
> Appendix D provides systematic experiments and analysis of different pre-trained VLM critic designs. We evaluated four distinct pre-training strategies (see Table 7), comparing their performance and design motivations to validate the effectiveness of the key components. Thank you!
>
>
>
> **Q4**
>
> It is difficult for prompts to strictly assess what is wrong. We validated the robustness of our method through the RSF module (which simulates VLM action uncertainty via randomness injection) and systematically evaluated ERPV under varying VLM qualities (including worst-case scenarios), as shown in Table 4. All results confirm that ERPV remains robust across settings, with performance consistently not falling below that of vanilla RL. Thank you!
>
>
>
> **Q5**
>
> We now focus on actions rather than state representations or reward shaping because actions are the only variables directly interacting with the environment and can directly calculate Q to reflect the value of the action, thereby more effectively influencing the exploration of RL. We will further expand our work to other multimodal feedback forms in the future. Thank you!

---

### Official Review · Reviewer_oTMT · 2025-11-01

**Soundness:** 2
**Presentation:** 3
**Contribution:** 2
**Rating:** 4
**Confidence:** 4

**Summary:**

This paper proposes ERPV, to address the issue of low exploration efficiency in VRL by introducing prior knowledge provided by pre trained VLMs. Experiments are conducted on Carla, DMC and CarRacing.

**Strengths:**

1. To the best of my knowledge, the paper is the first to systematically identify and formalize the challenge of “partially reliable knowledge.”

2. The work targets the central tension in integrating VLMs with VRL.

3. The experimental evaluation is extensive.

**Weaknesses:**

1. The core idea does not move beyond the teacher–student paradigm and remains within the traditional setting where the teacher provides knowledge and the student learns it.

2. Discarding the VLM at test time forfeits substantial information encoded in the VLM and risks overfitting to the test environment due to limited training.

3. There is no comparison of computational cost or runtime for the training phase.

**Questions:**

1. Given that VPG can already approximate the ground-truth Q reasonably well, why not directly use this Q estimate to optimize the actor network?

2. In VER, high consistency between the VLM and RL policies indicates reliability, whereas low consistency indicates unreliability and triggers increased exploration. Suppose the VLM’s action is unreliable, but after exploration the RL policy becomes reliable and efficient; then the consistency between the VLM and RL policies would remain low. In that case, would ERPV continue to explore such states indefinitely? Furthermore, if both the VLM’s action and the RL’s action are reliable but diverse, the consistency may still remain low. In this scenario, does ERPV favor exploration or exploitation?

---

> ### Author Response · Authors · 2025-11-23
>
> Thank you very much for your time and effort in reviewing our work. We truly appreciate your constructive and insightful feedback. Below, we provide detailed responses to the **Weaknesses** raised in your review:
>
> **W1**
>
> ERPV is based on the teacher-student paradigm, with the key innovation being the use of a **partially** reliable VLM to guide RL. Unlike traditional methods (e.g., [1, 2]) that assume the teacher is fully reliable, our approach effectively addresses the unreliability of the VLM.
>
>
>
> **W2**
>
> Our method focuses on real-time control, so VLM is not available in the test. To evaluate overfitting, we tested the model weights trained in the HW scenario in the following two situations:
>
> 1.For different weather tests, we changed the weather in the HW scene to rain to modify the visual input.
>
> | Scenario        | HW                       | HW with rain          |
> | --------------- | ------------------------ | --------------------- |
> | Model \ Metrics | ER, DD                   | ER, DD                |
> | DeepMDP         | 86 ± 43, 105 ± 50        | 2 ± 3, 14 ± 5         |
> | ERPV            | **299 ± 106, 311 ± 109** | **83 ± 38, 129 ± 52** |
>
> 2.Cross-scenario testing.
>
> | Scenario        | HW                       | JW                   |
> | --------------- | ------------------------ | -------------------- |
> | Model \ Metrics | ER, DD                   | ER, DD               |
> | DeepMDP         | 86 ± 43, 105 ± 50        | 9 ± 2, 10  ± 2       |
> | ERPV            | **299 ± 106, 311 ± 109** | **37 ± 12  38 ± 12** |
>
> The results show that it still outperforms the basic RL, confirming that there is no excessive overfitting.
>
>
>
> **W3**
>
> The graphics card used in this paper is an 80GB PCIe NVIDIA A100. Section E.4 of Appendix presents the GPU memory usage during EPRV training with different large models. The more comprehensive training cost for different methods is shown in the following table:
>
> | Method          | train time (hours) | Memory (M) | ER, DD                    |
> | --------------- | ------------------ | ---------- | ------------------------- |
> | DSF [3]         | $\approx $ 34.3              | $\approx $20971     | 208 ± 82，220 ± 86        |
> | ASF [1]         | $\approx $ 25.9              | $\approx $20971     | 207 ± 86， 221 ± 90       |
> | Vanilla SAC [4] | $\approx $ 8.3               | $\approx $3366      | 86 ± 43, 105 ± 50         |
> | ERPV            | $\approx $ 26.6              | $\approx $ 20971     | **299 ± 106， 311 ± 109** |
>
> RL combined with VLM will definitely slow down, but ERPV performs better than other methods. We would revise the content of Section E.4 of Appendix to more comprehensively compare the training cost of the training process. Thank you!
>
>
>
> **References**
>
> [1] Zihao Zhou, Bin Hu, Chenyang Zhao, Pu Zhang, and Bin Liu. Large language model as a policy teacher for training reinforcement learning agents. In Proceedings of the Thirty-Third International Joint Conference on Artificial Intelligence, pp. 5671–5679, 2024.
>
> [2] Donghoon Lee, Tung M Luu, Younghwan Lee, and Chang D Yoo. Sample efficient reinforcement learning via large vision language model distillation. In ICASSP 2025-2025 IEEE International Conference on Acoustics, Speech and Signal Processing (ICASSP), pp. 1–5. IEEE, 2025.
>
> [3] Yi Xu, Yuxin Hu, Zaiwei Zhang, Gregory P Meyer, Siva Karthik Mustikovela, Siddhartha Srinivasa, Eric M Wolff, and Xin Huang. Vlm-ad: End-to-end autonomous driving through vision-language model supervision. *arXiv preprint arXiv:2412.14446*, 2024b.
>
> [4] Tuomas Haarnoja, Aurick Zhou, Kristian Hartikainen, George Tucker, Sehoon Ha, Jie Tan, Vikash Kumar, Henry Zhu, Abhishek Gupta, Pieter Abbeel, et al. Soft actor-critic algorithms and applications. *arXiv preprint arXiv:1812.05905*, 2018b.

---

> ### Author Response · Authors · 2025-11-23
>
> Thank you very much for your time and effort in reviewing our work. Below, we provide detailed responses to the **Questions** raised in your review:
>
> **Q1**
>
> Do you mean $Q_{r}$? $Q_{r}$ is actually used to optimize the actor network because it directly evaluates the actions of RL. $Q_{v}$ is the state-action value of the VLM. $Q_{f}$ is the difference between $Q_{r}$ and $Q_{v}$, and can only be used to calculate the weight of the loss.
>
>
>
> **Q2**
>
> (1) Regarding the first situation you mentioned, ERPV does not explore indefinitely for the following reasons:
>
> 1.The entropy coefficient $\alpha$ is constrained by an auxiliary regularization term (Eq. 7). We do not enforce excessive exploration and preserve the baseline exploration level of vanilla SAC.
>
> 2.The target $\bar{\alpha}$ introduced in Eq. 7 is bounded, inherently preventing unbounded exploration. Fig. 5(b) shows that $\alpha$ remains stable throughout training.
>
> 3.To further demonstrate robustness under highly unreliable VLM guidance, we report the change of $\alpha$ on the Cheetah-Run task in Table 2, where the VLM performs poorly (reward: 5 ± 3):
>
> |    Train Step    | SAC   | DeepMDP | ERPV  |
> | ------ | ----- | ------- | ----- |
> | 0      | 0.1   | 0.1     | 0.1   |
> | 10000  | 0.042 | 0.062   | 0.066 |
> | 30000  | 0.020 | 0.052   | 0.073 |
> | 60000  | 0.036 | 0.056   | 0.069 |
> | 80000  | 0.039 | 0.055   | 0.064 |
> | 100000 | 0.038 | 0.052   | 0.060 |
>
> Even in this case, $\alpha$ fluctuates within a reasonable range without diverging.
>
>
>
> (2) As for the second situation: while ERPV does encourage exploitation when multiple actions are reliable, such a situation is rare in practice. The reason is that, under the influence of the VPG module, the RL policy is guided toward the VLM-provided reliable actions, causing their values to align in some states and yielding higher cumulative rewards rather than maintaining high action diversity.

---

### Meta-Review · Area_Chair_2fPk · 2026-01-06

**Summary:**

The paper explores the idea of leveraging prior knowledge contained in vision language models (VLMs) to guide RL agents in visual domains. Given the potentially unreliable nature of VLMs in these high dimensional domains, the authors introduce Value-aware Policy Guidance, which compares the VLM actions to that of the Q values, and VLM-guided Entropy Regularization, which adjust the entropy coefficient on the consistency between the VLM and the RL agent.

The idea to inject the VLM's pretrained knowledge into RL agents trained from scratch is an important idea that has been studied in the past few years. Some reviewers mention the concern of the computational cost of using VLMs during RL training. This is a fair point, but if the resulting agent is stronger and VLMs are not used at inference time, then it is acceptable. Reviewer cNdj mentioned doubts about the reliability of the VLM in outputting actions. This is a very important concern. If we look at the tables and try to understand what's the standalone performance of the VLM on these tasks, the closest thing we can look into is VBE. On DMControl, this performance is essentially random. It brings the question, how is it even possible for an essentially random agent to guide an RL agent and surpass its base performance? Looking at the different tables, ERPV numbers are consistently marked as bold, even though the confidence interval overlaps with other baselines. Additionally, some newer baselines are completely disregarded (e.g. DrQv2). Overall, these do not induce confidence in the reader. Looking deeper into the paper, it seems all reviewers have missed the fact that the critic used to select between VLM actions and RL actions is itself a pretrained network on the environments being tested. This is somewhat referred to by Reviewer K2Jp. The only place where this detail is clearly described in the appendix. Overall, given all these different concerns, doubts in the ability for VLMs to, at times, output useful actions for continuous control are not resolved.

**Reviewer Concerns:**

Reviewer K2Jp had concenrs about the training details, such as hyperparameter and the reliance on a pretrained critic. The concern about hyperparameter detail could be resolved, but not the one about the assumption of a pretrained critic. The concerns about Reviewer oTMT and ZmUU on the computational cost were clearly answered. I think this can likely remain a sticking point, even if the VLM is not used at inference time. Reviewer ZmUU additionally had concerns about the theoretical justification. Finally, Reviewer cNdj expressed doubt that it could work on very high dimensional environments. Results on quadruped do show some good results, but in the light of the previous experimental irregularities, it is not clear if those numbers can be trusted.

**Reviewer Scores:**

It is unlikely that any of the reviewers would change their score, except for perhaps reviewer ZmUU.

---

### Decision · Program_Chairs · 2026-01-26

Reject